# Mitigating Mask Prior Drift and Positional Attention Collapse in Large Diffusion Vision-Language Models

**Sujung Hong** [1]  **Chanyong Yoon** [1]  **Seong Jae Hwang** [1]

 hong-sujung/MPD-PAC     Project Page

## Abstract

Large diffusion vision–language models (LD-VLMs) have recently emerged as a promising alternative to autoregressive models, enabling parallel decoding for efficient inference and leveraging bidirectional attention for global context. Despite these advances, their behavior under long-form generation remains underexplored. In this work, we show that existing LDVLMs suffer from repetitive generation and degraded visual grounding, and identify two underlying causes. First, repetitive generation originates from a mask token prior: since generation tokens are initialized as mask tokens, their hidden representations progressively drift toward a shared prior direction over generation steps. Second, a fundamental misalignment between the positional attention bias and the iterative unmasking process suppresses attention toward informative visual tokens, degrading visual grounding. Based on these insights, we propose a training-free approach, introducing Mask Prior Suppression and Monotonic RoPE Scaling to mitigate mask prior drift and positional attention collapse during decoding. Experiments on general multimodal benchmarks and visual grounding tasks demonstrate improvements over baseline LDVLMs, with robust gains on long-form description benchmarks. Our results show that these failures can be effectively addressed with a lightweight, plug-and-play strategy that requires no additional training and generalizes across diverse LDVLM architectures.

[1]Department of Artificial Intelligence, Yonsei University, Seoul, Republic of Korea. Correspondence to: Seong Jae Hwang <seongjae@yonsei.ac.kr>.

*Proceedings of the 43rd International Conference on Machine Learning*, Seoul, South Korea. PMLR 306, 2026. Copyright 2026 by the author(s).

## 1. Introduction

Autoregressive large language models (LLMs) have become the dominant paradigm for reasoning and natural language generation (Brown et al., 2020; Radford et al., 2019; Dubey et al., 2024; Touvron et al., 2023; Peng et al., 2023; Wei et al., 2022). However, their sequential decoding limits inference efficiency and prevents refinement (Khoshnoodi et al., 2024). To address these limitations, large language diffusion models (LLDMs) (Ye et al., 2025; Nie et al., 2025b; Li et al., 2025c) have emerged as a promising alternative. Building upon this paradigm, diffusion-based generation has been extended to the vision-language domain, resulting in large diffusion vision-language models (LDVLMs) (Li et al., 2025b; Yang et al., 2025; You et al., 2025). These models enable globally consistent multimodal reasoning by maintaining interactions between visual and textual tokens throughout the generation process.

LDVLMs exhibit several distinctive advantages over autoregressive multimodal models. First, LDVLMs enable explicit control over the trade-off between generation quality and inference latency by adjusting the number of generation steps. They employ an iterative unmasking process, in which all generation tokens are initialized from a shared mask token $\mathcal{M}$ and progressively refined into meaningful tokens through a fixed number of generation steps. Second, LDVLMs facilitate global context modeling and structured output formation, which are critical for complex vision–language tasks. They adopt a bidirectional unmasking mechanism that allows each token to attend to all other tokens at every generation step, in contrast to the causal attention imposed by autoregressive models. To support stable bidirectional attention, LDVLMs typically employ Rotary Position Embeddings (RoPE) (Su et al., 2024), which encode relative positional information while preserving access to global context.

Despite these strengths, the behavior of LDVLMs in complex multimodal settings remains insufficiently explored. In this work, we identify two fundamental challenges that arise during long-form multimodal generation, as illustrated in Figure 1. First, we observe a persistent repetition of spe-

cific tokens during decoding that is largely independent of the input text. This phenomenon becomes increasingly severe as the number of generation steps decreases. We refer to this behavior as *mask prior drift*. Because all generation tokens are initialized from the same mask token $\mathcal{M}$, their hidden representations progressively converge toward a shared prior direction. This convergence limits semantic diversity across tokens and ultimately leads to repetitive generation. Second, we find that LDVLMs suffer from degraded visual grounding in long-form generation. Our analysis attributes this issue to positional attention collapse, induced by the inherent locality bias of RoPE. During iterative unmasking, attention concentrates on nearby semantically incomplete mask tokens, suppressing attention to distant informative visual tokens. Together, these issues weaken visual–textual alignment and limit the reliability of LDVLMs in producing high-quality long-form descriptions.

To address these issues, we propose two inference-time techniques: Mask Prior Suppression and Monotonic RoPE Scaling. First, Mask Prior Suppression reduces representational drift by adjusting the final hidden states of generation tokens in the mask prior subspace. This operation effectively reduces structured repetition while preserving semantic expressiveness. Second, we introduce Monotonic RoPE Scaling to improve positional attention stability. By monotonically emphasizing low-frequency RoPE components, generation tokens maintain stable attention to distant visual tokens and preserve RoPE's relative distance structure. Both techniques operate entirely at inference time and require no parameter updates or retraining.

Our main contributions are summarized as follows. First, we present a systematic analysis of the stability of the generation in LDVLMs, revealing repetitive patterns and degraded visual grounding as significant challenges. Second, we propose two training-free techniques, Mask Prior Suppression and Monotonic RoPE Scaling, that directly address the representational and positional biases underlying these failures. Finally, extensive experiments across diverse benchmarks and LDVLM architectures demonstrate consistent improvements in visual grounding and long-form generation quality, without retraining or additional parameters. Together, our methods provide a principled and lightweight solution for stabilizing long-form multimodal generation in LDVLMs.

## 2. Preliminaries

### 2.1. Large Diffusion Language Model

Diffusion Large Language Models (DLLMs) generate text via an iterative denoising procedure with parallel decoding, offering an alternative to autoregressive generation. Among these methods (Austin et al., 2021; Sahoo et al., 2024; Lou et al., 2024; Gong et al., 2025; Nie et al., 2025a;b; Ye et al., 2025), Masked Diffusion Models (MDMs) have emerged as

Please describe the image in detail.

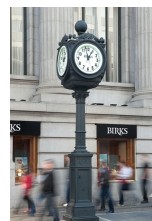

A prominent clock clock tower is situated in the center, the focal point of the image, and it a occupies a significant portion of the scene. The clock tower is vertically tall, stretching almost from the the left to the the right of the frame, extending from the the middle to the the bottom of the image.

Token Repetition: ▨    Visual Grounding Degradation: ▨

*Figure 1.* **Failure case of LDVLMs.** Under parallel decoding with 64 generation tokens and 16 generation steps, LLaDA-V produces highly repetitive phrases as highlighted in red, and exhibits degraded visual grounding as highlighted in gray.

a representative framework for discrete sequence modeling. MDMs assume an input sequence $x_0 = [x^i]_{i=1}^N$ consisting of $N$ tokens, including special mask tokens $\mathcal{M}$. The model defines the model distribution $p_\theta(x_0)$ through a forward and reverse diffusion process. In the forward process, $x_0$ is independently replaced by $\mathcal{M}$ with the time step $t$ uniformly sampled from the interval $[0, 1]$. The reverse process starts from a fully masked sequence and iteratively replaces $\mathcal{M}$ tokens to reconstruct the original sentence by sampling token values at masked positions from the conditional distribution $p_\theta(x_0 \mid x_t)$. During inference, generation proceeds for a fixed number of reverse steps $T$, where all masked positions are updated in parallel at each step. The number of steps $T$ governs the trade-off between decoding efficiency and generation quality.

### 2.2. Rotary Position Embedding

Rotary Position Embedding (RoPE) (Su et al., 2024) is a widely used mechanism for encoding relative positional information in transformer-based models. Let $x_m \in \mathbb{R}^d$ denote the $d$-dimensional embedding vector at position $m$, where $d$ is even. The query and key representations are obtained via linear projections $q_m = W_Q x_m$ and $k_n = W_K x_n$, where $W_Q, W_K \in \mathbb{R}^{d \times d}$. RoPE encodes positional information by partitioning the embedding dimensions into $d/2$ two-dimensional subspaces. Each subspace is associated with a frequency $\theta_i = 10000^{-2i/d}$ for $i \in \{0, 1, \ldots, d/2 - 1\}$, and applies a position-dependent rotation

$$r(\theta_i, m) = \begin{pmatrix} \cos(m\theta_i) & -\sin(m\theta_i) \\ \sin(m\theta_i) & \cos(m\theta_i) \end{pmatrix}.$$

Stacking all subspaces yields a block-diagonal rotation matrix $R_{\Theta,m}^d = \mathrm{diag}\big(r(\theta_0, m), \ldots, r(\theta_{d/2-1}, m)\big)$, which is applied to the query and key vectors as $q_m' = R_{\Theta,m}^d q_m$ and $k_n' = R_{\Theta,n}^d k_n$. A key property of RoPE is that the resulting attention score depends only on the relative position between tokens, i.e., $q_m'^\top k_n' = q_m^\top R_{\Theta,n-m}^d k_n$. Thus, RoPE implicitly encodes relative positional information through rotational offsets in the query–key interaction.

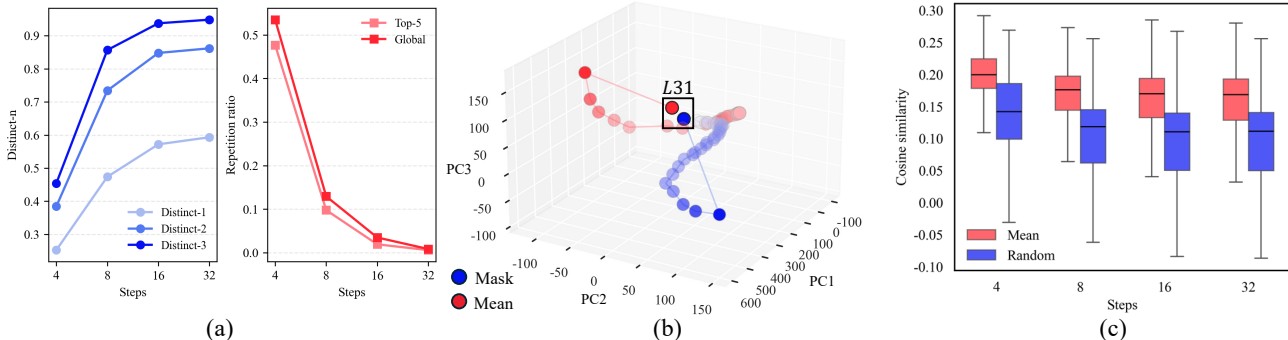

*Figure 2.* **Visualization of token repetition and mask prior drift.** (a) Distinct-$n$ (left) and repetition ratio (right) across different numbers of generation steps. Fewer generation steps lead to lower distinct-$n$ and higher repetition. (b) 3D PCA trajectories of hidden states for the vocabulary mean embedding and the uncontextualized mask token, which converge to a similar region at the final layer ($L31$). (c) Cosine similarity between contextualized mask token embeddings and the vocabulary mean, showing consistently stronger alignment than random embeddings, especially with fewer generation steps.

## 3. Related Work

### 3.1. Large Diffusion Vision–Language Model

DLLMs have recently been extended to the vision–language domain. Previous studies mainly focus on designing training frameworks with multimodal data (You et al., 2025; Li et al., 2025b; Yang et al., 2025). For instance, LaViDa (Li et al., 2025b) introduces complementary masking, which ensures that all tokens are considered across masking patterns during training. Similarly, MMaDA (Yang et al., 2025) and Lumina-DiMOO (Xin et al., 2025) proposes a unified architecture supporting textual reasoning, multimodal reasoning, and text-to-image generation. A complementary line of work targets the decoding process itself, for example by reducing redundant suffix-mask attention (Chen et al., 2025) or by restructuring the decoding paradigm into block-wise variants (Wang et al., 2025c). These directions modify the structural inputs or schedule of unmasking, whereas our method operates at the representation and positional levels and is largely orthogonal to such structural interventions; we provide a direct empirical comparison in Section E. Despite these advances, prior work has paid limited attention to intrinsic generation characteristics in LDVLMs for visual–textual alignment.

### 3.2. Rotary Position Embeddings in VLMs

Rotary Position Embedding (RoPE) has been extended to multimodal models through two broad approaches. The first approach applies the one-dimensional positional mechanism originally designed for text tokens to visual tokens, often combined with dynamic position scaling to handle large positional indices (Su et al., 2024; Ge et al., 2025). The second approach assigns distinct frequency components to different spatial or temporal dimensions to better capture multimodal structure (Wang et al., 2024; Wei et al., 2025; Li et al., 2025a). In addition, CircleRoPE (Wang et al., 2025a) arranges visual tokens in a circular layout orthogonal to text, promoting uniform cross-modal attention. While prior work

extends positional encoding for multimodal autoregressive models, typically requiring training, positional mechanisms for MDMs with bidirectional attention and parallel unmasking remain largely unexplored.

## 4. Analysis of Generation Failures in LDVLMs

While LDVLMs demonstrate competitive performance relative to autoregressive models, they often suffer from repetitive generation patterns and suboptimal visual grounding. In this section, we investigate the causes of these behaviors by analyzing the underlying architectural design and hidden state dynamics.

### 4.1. Token Repetition and Mask Prior Bias

**Speed–quality trade-off and repetition.** LDVLMs exhibit an inherent trade-off between decoding speed and generation quality, which is closely related to the number of tokens unmasked at each iteration. As the number of generation steps decreases, a larger proportion of tokens is updated simultaneously. This parallel unmasking reduces the effective conditioning depth through which contextual information can be propagated across transformer layers. In practice, such behavior frequently manifests as consecutive token repetition during inference. Although this speed–quality trade-off has been widely observed, the underlying causes of quality degradation, particularly repetitive generation, remain insufficiently understood.

**Structured token repetition.** To analyze this phenomenon, we perform multimodal text generation on 100 randomly sampled images from the COCO 2014 validation set using the instruction "`Please describe the image in detail`." For each image, we generate 64 tokens while varying the number of generation steps $T \in \{4, 8, 16, 32\}$. We evaluate generation quality using two metrics: *distinct-$n$* and *repetition ratio*. Distinct-$n$ measures the proportion of unique $n$-grams to assess lexical diversity, while repetition ratio quantifies the fraction of consecutively repeated to-

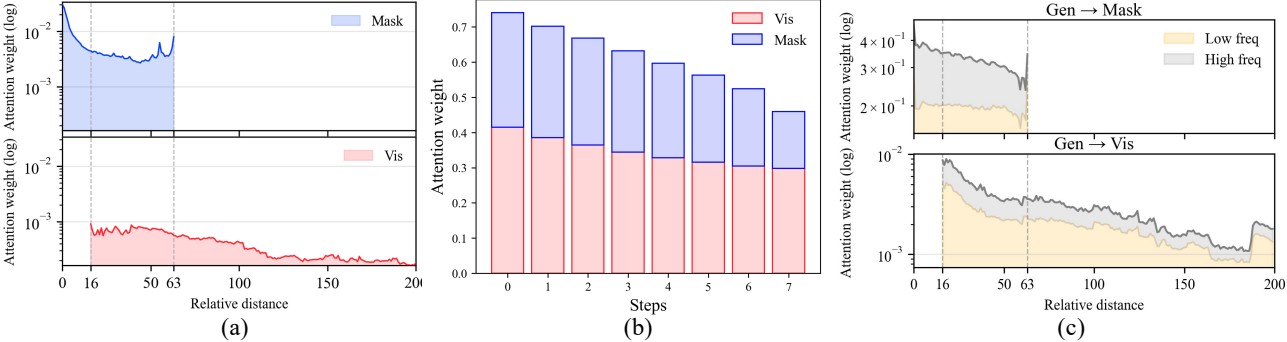

*Figure 3.* **Visualization of positional attention collapse.** (a) Mean attention weight (log scale) across relative distance, showing stronger attention to mask tokens than visual tokens at similar distances and an overall decreasing trend in attention to visual tokens as relative distance increases. (b) Sum of attention to visual and mask tokens per generation token across generation steps, revealing a persistent allocation of comparable attention weights to mask tokens despite their lack of semantic content. (c) Frequency decomposition of attention over relative distance, where RoPE dimensions are evenly divided into high- and low-frequency bands. High-frequency components dominate at short ranges, while low-frequency components dominate at long ranges, yet overall attention remains weak.

kens. As shown in Figure 2(a, left), decreasing the number of steps leads to a sharp decline in distinct-n. Importantly, we find that this repetition is highly structured rather than random, concentrating on a small set of function tokens. By analyzing the top-$k$ logit tokens of the uncontextualized mask token (e.g., \n, the, ,, <space>, .), we observe that these few tokens account for over 95% of all repeated outputs, as illustrated in Figure 2(a, right). Here, an *uncontextualized mask token* refers to the mask token processed without any surrounding text or visual input. This result suggests that as the number of generation steps decreases, the model fails to leverage contextual signals and instead collapses toward the prior encoded in the mask token.

**Mask prior drift in hidden states.** We hypothesize that this structured repetition originates from the shared initialization of all generation tokens as the mask token, which introduces a representation prior into the decoding process. The mask token is trained to represent a context-agnostic and uninformative state, and its embedding is therefore biased toward the mean of the vocabulary embedding space. Let $f_l(\cdot)$ denote the transformation induced by the $l$-th transformer layer, and let $h_l^{\mathcal{M}} = f_l(e_{\mathcal{M}})$ be the hidden state of the mask token at layer $l$. We define the vocabulary embedding mean as $\hat{e} = \frac{1}{|V|} \sum_{v \in V} e_v$, where $V$ denotes the set of vocabulary embeddings, and similarly obtain its layer-wise representations as $h_l^{\hat{e}} = f_l(\hat{e})$. As shown in Figure 2(b), a 3D PCA visualization of the joint set $\{h_l^{\mathcal{M}}\}_{l=1}^{L} \cup \{h_l^{\hat{e}}\}_{l=1}^{L}$, where $L$ denotes the total number of transformer layers, reveals that the hidden states $h_l^{\mathcal{M}}$ and $h_l^{\hat{e}}$ progressively align at the final transformer layer. While their trajectories remain closely aligned in the early layers and exhibit distinct characteristics in the intermediate layers, they converge toward a similar region in the final layer ($L31$).

**Mask prior drift during inference.** We further investigate whether this representation drift persists when mask tokens become contextualized during inference. Here, con-

textualized refers to tokens that are processed jointly with other visual and textual tokens as part of the model input. We use the same dataset and instructions as in the previous experiment and perform generation on 10 randomly sampled images, producing 64 tokens for each image. We find that contextualized mask token embeddings exhibit higher cosine similarity to the vocabulary mean embedding than randomly sampled token embeddings, as illustrated in Figure 2(c). This effect is exacerbated as the number of decoding steps decreases, since parallel decoding restricts conditioning signals and amplifies the bias of the mask token, ultimately leading to highly structured token repetition.

**Empirical scope of our claim.** We emphasize that this analysis is empirical: we do *not* claim that the vocabulary mean fully characterizes the nonlinear dynamics of mask-token collapse. Rather, motivated by the Linear Representation Hypothesis (Park et al., 2024), our claim is that there exists a dominant low-dimensional bias direction in the final hidden-state space, which is well approximated by the vocabulary mean and which is consistently aligned with contextualized mask states across decoding steps. A detailed comparison against alternative prior directions, including frequency-weighted lexical priors and random directions, and the stability of this alignment across generation steps, are provided in Section C.

### 4.2. Positional Attention Collapse

**Positional bias under iterative unmasking.** Beyond token repetition, LDVLMs often exhibit degraded visual grounding. This degradation arises from a structural misalignment between the locality bias induced by RoPE-based attention and the iterative unmasking process. Motivated by this observation, we analyze how RoPE interacts with the generation dynamics of LDVLMs. While the long-range decay property of RoPE (Su et al., 2024) facilitates coherence in autoregressive language modeling, it introduces a critical

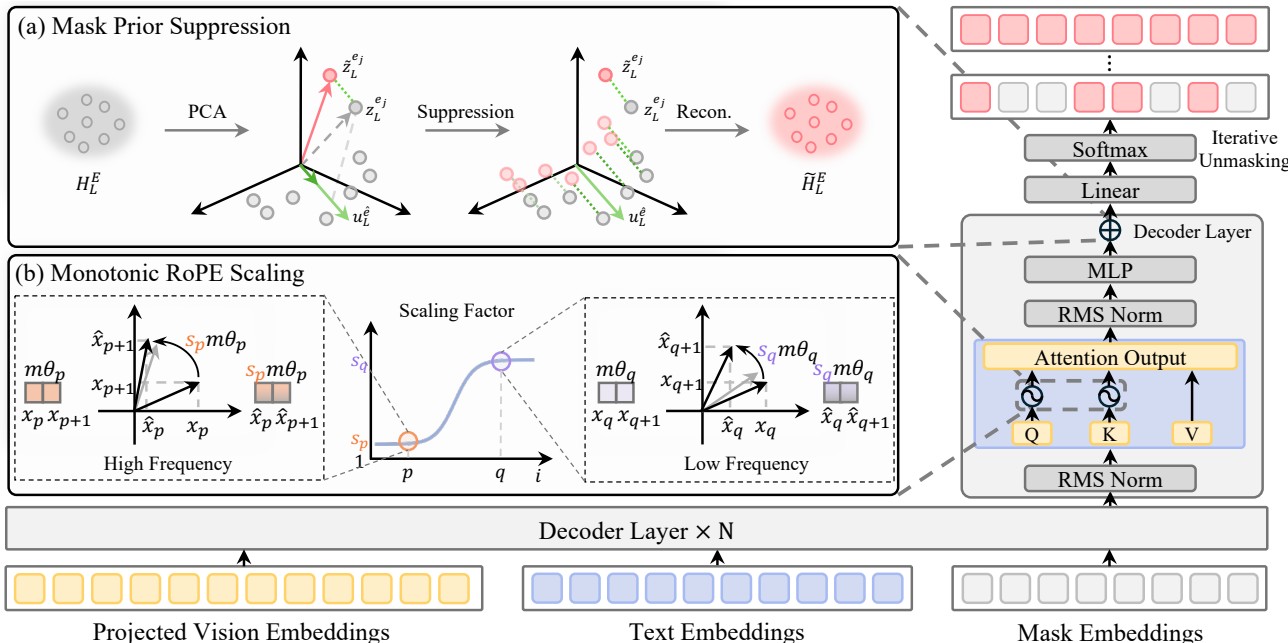

*Figure 4.* **Overview of the proposed model.** (a) *Mask prior suppression.* The final hidden state $h_L^{e_j}$ is decomposed along the prior direction $u_L^{\hat{e}}$, and prior components are adaptively suppressed based on cosine similarity. (b) *Monotonic RoPE scaling.* Low-frequency RoPE components, which govern long-range positional interactions, are scaled more strongly than high-frequency components to preserve attention to distant visual tokens, where $m$ denotes the token position.

bottleneck for VLMs, which require more global access to visual information (Ding et al., 2024; Wang et al., 2025a; Wei et al., 2025; Li et al., 2025a). This limitation is further exacerbated in LDVLMs, where generation proceeds via iterative unmasking under bidirectional attention. Generation tokens originate from semantically uninformative mask tokens. During iterative unmasking, close positional proximity leads to disproportionately high attention.

**Attention collapse toward mask tokens.** To investigate this behavior, we analyze attention distributions during generation. As illustrated in Figure 3(a), generation tokens allocate substantial attention weights to mask tokens whose semantic representations are not fully formed. Moreover, Figure 3(b) shows that mask tokens receive a total amount of attention comparable to that of visual tokens across generation steps. As a result, attention to distant visual information is suppressed, leading to degraded visual grounding.

**Frequency-wise decomposition of attention.** To further examine this structural limitation, we decompose RoPE-based attention into frequency components and examine their respective contributions to attention allocation. RoPE implicitly orders embedding dimensions by frequency, where each pair of dimensions is associated with a predefined frequency $\theta_i$: lower indices correspond to higher-frequency components, while higher indices capture lower-frequency components. As shown in Figure 3(c), attention from generation tokens to mask tokens (Gen→Mask) is dominated by high-frequency components. This dominance induces

strong local attention concentration among generation tokens. In contrast, attention from generation tokens to visual tokens (Gen→Vis) increasingly relies on low-frequency components as relative distance grows, reflecting their role in supporting long-range interactions. However, despite this shift, overall attention still decays with distance, leaving distant visual tokens weakly attended. Overall, these results indicate that while low-frequency components partially support long-range attention, RoPE's frequency-dependent behavior ultimately biases attention toward nearby generation tokens, limiting effective visual grounding.

## 5. Method

Motivated by the analysis in Section 4, we propose two training-free approaches to mitigate the mask prior drift and attention collapse identified in LDVLMs. As illustrated in Figure 4, we first introduce a *Mask Prior Suppression* mechanism to debias the inherent prior in mask embedding hidden states. Subsequently, we employ *Monotonic RoPE Scaling* to alleviate visual grounding degradation arising from vision-language misalignment.

### 5.1. Mask Prior Suppression

Our analysis reveals that mask token representations tend to drift toward the vocabulary mean in the final hidden states. To address this issue, we apply a prior suppression mechanism to the final hidden states. Let $E = (e_1, \ldots, e_J)$ denote an input embedding sequence of length $J$, where $e_j$

is the embedding of the $j$-th token. Let $H_l^E = f_l(E) = (h_l^{e_1}, \ldots, h_l^{e_J})$ denote the corresponding hidden states at the $l$-th transformer layer, where $h_l^{e_j}$ is the hidden state corresponding to the $j$-th embedding at layer $l$. We further denote by $\mathcal{S}_{\mathcal{M}} \subseteq \{1, \ldots, J\}$ the set of token positions occupied by mask tokens, i.e., $j \in \mathcal{S}_{\mathcal{M}}$ iff $e_j = e_{\mathcal{M}}$.

**Prior subspace construction.** To capture the intrinsic prior geometry induced by an uncontextualized masked token, we approximate the mask token prior using the mean vocabulary embedding $\hat{e}$, which represents a context-agnostic average over token semantics. This choice is motivated by the Linear Representation Hypothesis (Park et al., 2024), which suggests that even in nonlinear models, meaningful concepts can emerge as approximately linear directions in representation space. We empirically verify in Section C that the vocabulary mean is highly aligned with both a frequency-weighted lexical prior and contextualized mask-token states, while a random direction is not, supporting its use as a stable, parameter-free proxy for the dominant collapse direction. We thus forward $\hat{e}$ through the transformer layers to obtain a set of layer-wise mean representations $\{h_l^{\hat{e}}\}_{l=1}^{L}$. We then compute their mean $\mu = \frac{1}{L} \sum_{l=1}^{L} h_l^{\hat{e}}$ and perform PCA on the centered representations, yielding an orthonormal basis $U \in \mathbb{R}^{d \times k}$ that spans the mean prior subspace, where $k$ denotes the number of principal components.

**Projection and similarity estimation.** For each token $e_j$, we define its deviation in the final layer as $\delta_L^{e_j} = h_L^{e_j} - \mu$. This deviation is projected onto the prior subspace: $z_L^{e_j} = U^\top \delta_L^{e_j} \in \mathbb{R}^k$. Similarly, the projected prior direction is denoted as $z_L^{\hat{e}} = U^\top (h_L^{\hat{e}} - \mu)$ with its normalized unit vector $u_L^{\hat{e}} = \frac{z_L^{\hat{e}}}{\|z_L^{\hat{e}}\|_2}$. We quantify the alignment between a token and the mask prior using cosine similarity:

$$c^{e_j} = \frac{\langle z_L^{e_j}, u_L^{\hat{e}} \rangle}{\|z_L^{e_j}\|_2}.$$

**Adaptive prior suppression.** We attenuate components that are aligned with the prior direction in proportion to this similarity. Let $\alpha^{e_j} = \lambda \max(0, c^{e_j})$ be the suppression strength, where $\lambda$ is a positive hyperparameter. The updated projection is computed as:

$$\tilde{z}_L^{e_j} = z_L^{e_j} - \alpha^{e_j} \langle z_L^{e_j}, u_L^{\hat{e}} \rangle u_L^{\hat{e}}.$$

The modified projection $\tilde{z}_L^{e_j}$ is then mapped back to the original feature space as $\tilde{h}_L^{e_j} = h_L^{e_j} + U(\tilde{z}_L^{e_j} - z_L^{e_j})$, which selectively removes only the component associated with the mask prior. This operation is applied to all masked positions, i.e., $\forall j \in \mathcal{S}_{\mathcal{M}}$, effectively preserving the semantic integrity of unmasked tokens while preventing representational collapse in the generated sequence. Therefore, by applying this operation to all masked positions, we obtain the updated final hidden states $\tilde{H}_L^E$ from the original $H_L^E$, as illustrated in Figure 4(a).

**Remark on intervention scale.** Mask Prior Suppression modifies only $k = 3$ out of $d = 4096$ dimensions ($\sim 0.07\%$) at the final layer, comparable in scale to representation-level interventions used in autoregressive LLMs without retraining (Li et al., 2023; Wang et al., 2025b). This small footprint, combined with the consistent gains we observe across multiple LDVLM architectures and benchmarks (Tables 1 and 11), suggests that the modification does not broadly disrupt the learned diffusion dynamics.

## 5.2. Monotonic RoPE Scaling

We introduce a frequency-aware adjustment to RoPE that applies monotonic scaling across frequency components. Rather than uniformly scaling all RoPE frequencies, our method progressively modulates the scaling strength according to the frequency index.

**RoPE scaling function.** We first normalize the frequency index $i \in \{0, \ldots, d/2 - 1\}$ as $\tau_i = i/(d/2 - 1)$. Recall that $\theta_i = 10000^{-2i/d}$, so larger $i$ corresponds to a lower RoPE frequency. We define a sigmoid gate

$$\gamma_i = \sigma\big(\eta(\tau_i - \tau_0)\big), \qquad \sigma(\xi) = \frac{1}{1 + \exp(-\xi)},$$

where $\tau_0$ specifies the frequency boundary at which scaling becomes active and $\eta > 0$ controls the sharpness of the transition. The scaling factor is then given by

$$s_i = 1 + \beta\, \gamma_i,$$

where $\beta \geq 0$ controls the maximum scaling.

**Frequency-aware modulation.** This formulation induces a smooth and monotonic scaling schedule along the frequency axis. High-frequency components are assigned scaling factors close to 1, while lower-frequency components are progressively amplified, with the overall scaling smoothly bounded within $(1, 1 + \beta)$. We apply frequency-aware scaling to the RoPE rotation angle as

$$m\theta_i \;\leftarrow\; s_i\, m\theta_i.$$

As a result, high-frequency positional information is largely preserved, whereas lower-frequency components receive stronger scaling, which prevents attention collapse at large positional distances, as shown in Figure 4(b).

## 6. Experiment

**Implementation details.** We implement our proposed method on top of LLaDA-V (You et al., 2025) and LaViDa (Li et al., 2025b), and observe consistent trends across both backbones. Additional analyses for each model are provided in Section H. For the LLaDA-V-based implementation, we set the hyperparameters to $\lambda = 0.1$, $\beta = 0.01$, and $\eta = 8.0$, while for LaViDa, we use $\lambda = 0.3$, $\beta = 0.01$, and $\eta = 12.0$. We fix $\tau_0 = 0.6$ and $k = 3$

*Table 1.* Comparison of LDVLMs on general benchmarks, visual grounding, and long-form generation tasks. **Bold**: best, underline: second best. AR: Autoregressive models, Diff.: LDVLMs.

| Model | Type | General | | | Visual Grounding | | | Long-form Generation | | |
|---|---|---|---|---|---|---|---|---|---|---|
| | | MME (sum)↑ | MMBench↑ | MMMU↑ | RefCOCOg↑ | Ferret↑ | GQA↑ | LLaVA-Bench↑ | DetailCaps↑ | MIA↑ |
| LLaVA-OV | AR | 1998 | 80.8 | 48.8 | 56.9 | 36.0 | 72.2 | 83.8 | 60.6 | 76.1 |
| Qwen2.5 | AR | 2448 | 83.5 | 58.6 | 16.6 | 53.8 | 70.6 | 96.6 | 63.3 | 84.6 |
| InternVL3 | AR | 2415 | 83.4 | 53.6 | 67.1 | 59.0 | 71.2 | 82.3 | 64.4 | 82.3 |
| LLaVA-1.6 | AR | 1842 | 68.1 | 42.6 | 28.1 | 60.3 | **75.8** | **75.3** | 60.6 | **76.1** |
| LLaDA-V | Diff. | 1998 | 82.9 | 48.6 | 64.8 | 60.4 | 61.6 | 61.3 | 59.8 | 66.1 |
| + Ours | Diff. | **2003** | **83.3** | **49.3** | **65.0** | **62.9** | 61.6 | 64.1 | **63.6** | 67.0 |
| LaViDa | Diff. | 1682 | 71.7 | 43.2 | 36.9 | 25.9 | 59.4 | 39.5 | 8.3 | 49.4 |
| + Ours | Diff. | 1705 | 72.0 | 43.7 | 44.0 | 35.7 | 60.2 | 46.5 | 56.1 | 57.3 |

Provide a short description for this region.

Imagine you are writing a short real estate advertisement for this room, mention three selling points without referring to the size or color of the room.

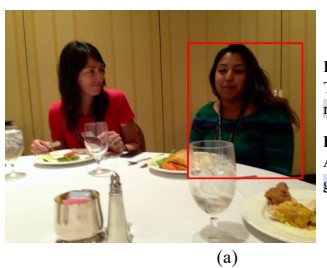

**LLaDA-V:**
The woman in the red shirt

**LLaDA-V+Ours:**
A woman in a green shirt

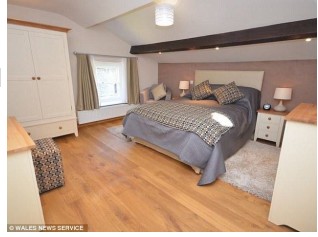

**LLaDA-V:**
This this room bedroom offers a cozy retreat inviting relaxation.1. The. The. The. The. The. 2.. The. The. The. The. The. The 3.. The. The room's design combines functionality and comfort, making it a perfect space for rest and unwind

**LLaDA-V+Ours:**
This room bedroom offers a cozy retreat with a comfortable and inviting bed. The soft pillows on the bed create a relaxing ambiance,, while the warm lighting from the lamps provides a cozy glow throughout the room. The wooden furniture adds a touch of warmth and style, making it a perfect space to relax and unwind

(a)       (b)

*Figure 5.* **Qualitative comparison on visual grounding and long-form generation.** (a) RefCOCOg results. A red box indicates the target region. The baseline model LLaDA-V produces descriptions referring to an incorrect object shown in gray, whereas our method correctly grounds the description to the target location and achieves more accurate visual grounding shown in blue. (b) MIA results. The baseline model exhibits repetitive descriptions shown in gray. Our method shown in blue effectively mitigates repetition and generates more coherent and informative long-form descriptions.

throughout all experiments. All hyperparameters are fixed across tasks and datasets for each backbone. To ensure a fair comparison with existing works, all evaluations are conducted using the LMMs-Eval (Zhang et al., 2024) framework. Further hyperparameter sensitivity experiments are presented in Section I.4, and detailed evaluation setups are described in Section B.

**Datasets.** To comprehensively evaluate the general reasoning, visual grounding, and long-form generation capabilities of our model, we conduct experiments across nine widely recognized benchmarks. For general-purpose evaluation, we adopt MME (Fu et al., 2025), MMBench (Liu et al., 2024c), and MMMU (Yue et al., 2024). To assess visual grounding, we utilize RefCOCOg (Kazemzadeh et al., 2014), Ferret (You et al., 2024), and GQA (Hudson & Manning, 2019). Finally, for long-form generation, we use LLaVA-Bench (Liu et al., 2023), DetailCaps (Dong et al., 2024), and MIA (Qian et al., 2025). For computational efficiency, we adopt GQA Lite for GQA, and randomly sampled 100 instances for both DetailCaps and Ferret. Further details are provided in Section A.

### 6.1. Main Results

**Overall performance.** Table 1 summarizes the performance of our method across general reasoning, visual grounding, and long-form generation tasks. Our approach achieves comparable or superior performance across the evaluated benchmarks without requiring any additional training. We compare to LLaVA-1.6-7B (Liu et al., 2024a), which is trained with a comparable data scale and parameter count, and further include baselines of similar model size trained on larger datasets, including LLaVA-One-Vision-7B (Li et al., 2024), Qwen2.5-VL-7B (Bai et al., 2025), and InternVL-3-8B (Zhu et al., 2025). Additional details on the compared models are provided in Section B.1. For LLaDA-V, our method demonstrates clear improvements on visual grounding benchmarks such as RefCOCOg and Ferret, as well as on the long-form generation benchmark LLaVA-Bench and DetailCaps, while maintaining competitive performance on general reasoning tasks. A similar trend is observed for LaViDa, where grounding accuracy improves substantially alongside consistent gains in long-form generation quality. The degraded quality of the LaViDa baseline is further investigated through a detailed performance analysis in Section H.

**Robustness across generation steps.** As illustrated in Figure 6, our approach consistently produces coherent and diverse outputs across different generation settings. Due to the mask token prior, baseline LDVLMs tend to repeatedly generate mean embedding prior tokens, which often manifests as repeated generation of specific words. This behavior leads to premature termination through excessive

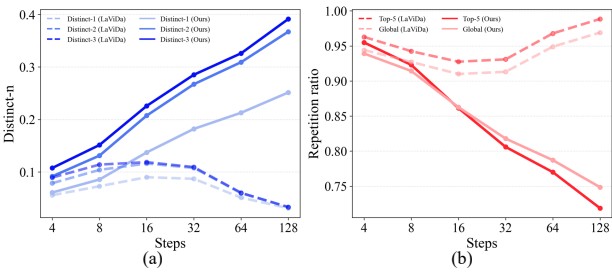

*Figure 6.* **Results of LaViDa on DetailCaps with varying generation steps.** Dashed lines: LaViDa, solid lines: Ours. (a) Distinct-$n$ scores of our method exhibit an increasing trend across the evaluated settings and remain consistently higher than those of the baseline. (b) The repetition ratio under our method shows a decreasing trend across the evaluated settings and remains consistently lower than that of the baseline.

*Table 2.* Ablation study on visual grounding and long-form generation benchmarks across LLaDA-V and LaViDa. **Bold**: best, underline: second best; MPS: Mask Prior Suppression; MRS: Monotonic RoPE Scaling.

| Model | Method | | Visual Grounding | | Long-form Generation | |
|---|---|---|---|---|---|---|
| | MPS | MRS | RefCOCOg | Ferret | LLaVA-Bench | DetailCaps |
| LLaDA-V | | | 64.8 | 60.4 | 61.3 | 59.8 |
| | ✓ | | 64.8 | 60.2 | 61.7 | 60.0 |
| | | ✓ | 64.8 | 60.6 | 63.9 | 60.0 |
| | ✓ | ✓ | **65.0** | **62.9** | 64.1 | **63.6** |
| LaViDa | | | 36.9 | 25.9 | 39.5 | 8.3 |
| | ✓ | | **44.0** | 35.4 | 42.3 | **56.1** |
| | | ✓ | 39.5 | 29.8 | 41.0 | 8.3 |
| | ✓ | ✓ | **44.0** | **35.7** | 46.5 | **56.1** |

emission of the |eot| token or results in a degraded lexical diversity in the generated descriptions. In contrast, our method effectively suppresses the mask token prior, mitigating repetitive generation and enabling more informative and diverse long-form outputs.

**Qualitative results.** Qualitative examples in Figure 5 illustrate that our method mitigates failures of LDVLMs. Additional results on unified vision–language architectures and qualitative examples are provided in Sections G and J.

### 6.2. Ablation Study

**Ablation on MPS and MRS.** We conduct an ablation study to analyze the effects of Mask Prior Suppression (MPS) and Monotonic RoPE Scaling (MRS) on long-form generation and visual grounding tasks, as summarized in Table 2. Overall, the full model exhibits the most stable performance across benchmarks, indicating the benefit of addressing both mask prior drift and positional attention bias. On LLaDA-V, removing either component results in clear performance drops across tasks, suggesting that MPS and MRS play complementary roles. In particular, removing MPS tends to have a larger impact on long-form generation benchmarks such as LLaVA-Bench and DetailCaps, while removing MRS more noticeably affects visual grounding performance on RefCOCOg and Ferret. For LaViDa, individual components

*Table 3.* Comparison with representative RoPE scaling baselines.

| Method | LLaDA-V | | | LaViDa | | |
|---|---|---|---|---|---|---|
| | RefCOCOg | Ferret | DetailCaps | RefCOCOg | Ferret | DetailCaps |
| Baseline | 64.8 | 60.4 | 59.8 | 36.9 | 25.9 | 8.3 |
| + NTK | 61.9 | 59.6 | 60.9 | 39.6 | 30.9 | 7.9 |
| + YaRN | 59.3 | 56.0 | 59.1 | 39.6 | 30.7 | 7.9 |
| + Ours | **65.0** | **62.9** | **63.6** | **44.0** | **35.7** | **56.1** |

can be competitive on certain benchmarks. For instance, removing MRS yields results comparable to those on Ref-COCOg and DetailCaps. Nevertheless, the full model provides the best overall performance across tasks, supporting the importance of combining both mechanisms for robust and balanced multimodal generation.

**Choice of mask prior direction.** We analyze the choice of mask prior direction while keeping Monotonic RoPE Scaling fixed, as shown in Table 4. The vocabulary mean direction consistently yields the largest and most stable gains across both visual grounding and long-form generation benchmarks. This indicates that, beyond the overall performance improvement introduced by RoPE scaling, effective mask prior suppression critically depends on principled prior direction selection rather than subtracting an arbitrary component. A comparison against a decoding-level baseline (DPad-style suffix dropping) is provided in Section E, and further results in Section I.

**Comparison with RoPE scaling baselines.** We further verify that conventional RoPE rescaling does not address the locality bias under iterative unmasking. Table 3 compares NTK-aware scaling (LocalLLaMA Community, 2023; Liu et al., 2024b) and YaRN (Peng et al., 2024) as drop-in replacements for our Monotonic RoPE Scaling. Both baselines, designed for long-context extrapolation, often degrade visual grounding, whereas our monotonic low-frequency amplification yields consistent gains across both backbones.

### 6.3. Result Analysis

**Effect of Mask Prior Suppression.** To quantify mask prior drift during decoding, we measure the cosine similarity between the contextualized last layer hidden states of mask tokens and the vocabulary mean prior. High similarity reflects increasing alignment with a shared prior direction, leading to reduced semantic diversity and repetitive generation. As illustrated in Figure 7(a), our method consistently reduces this cosine similarity across generation steps. This observation indicates that suppressing a low-dimensional prior subspace is sufficient to prevent mask prior drift during decoding, without modifying the full representational space. Consequently, the model produces less repetitive and more semantically diverse generations. Moreover, since mask prior suppression operates on a low-dimensional subspace, it introduces minimal inference overhead, as confirmed by the latency analysis in Section I.5.

*Table 4.* Ablation study on different mask prior directions. We compare vocabulary-mean prior (ours), top-$k$ token-based priors, and random directions on visual grounding and long-form generation benchmarks.

| Model | Type | Visual Grounding | | Long-form Generation | |
|---|---|---|---|---|---|
| | | RefCOCOg | Ferret | LLaVA- Bench | MIA |
| LLaDA-V | Top-1 | 64.4 | 61.9 | 60.8 | 66.4 |
| | Top-3 | 65.0 | 62.4 | 65.3 | 65.4 |
| | Random1 | 65.0 | 61.8 | 62.0 | 66.5 |
| | Random2 | 64.7 | 61.0 | 62.9 | 66.4 |
| | **Ours** | 65.0 | 62.9 | 64.1 | 67.0 |
| LaViDa | Top-1 | 44.0 | 34.8 | 44.5 | 57.1 |
| | Top-3 | 44.0 | 35.3 | 44.6 | 56.0 |
| | Random1 | 44.0 | 34.8 | 45.1 | 57.0 |
| | Random2 | 44.0 | 34.6 | 41.6 | 56.9 |
| | **Ours** | 44.0 | 35.7 | 46.5 | 57.3 |

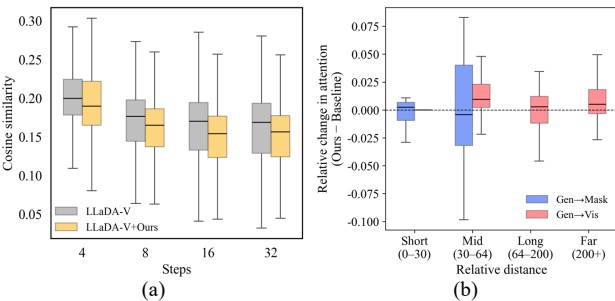

*Figure 7.* **Visualization of result analysis.** (a) Box plot of cosine similarity between contextualized mask tokens and the vocabulary mean, showing consistent reduction across generation steps. (b) Relative change in attention with respect to relative distance, where attention to distant visual tokens increases compared to the baseline, while attention to mask tokens is preserved or reduced.

**Effect of Monotonic RoPE Scaling.** To analyze how monotonic RoPE scaling affects positional attention, we examine changes in attention allocation with respect to relative token distance. Specifically, we distinguish attention to nearby mask tokens from attention to distant visual tokens, which is critical for visual grounding in LDVLMs. As shown in Figure 7(b), monotonic RoPE scaling gradually amplifies low-frequency components associated with long-range positional relations, leading to a redistribution of attention. Attention to nearby, semantically neutral mask tokens is reduced, while attention is reallocated toward distant visual tokens. By alleviating the misalignment between RoPE's locality bias and the bidirectional unmasking mechanism of LDVLMs, this adjustment preserves global access to visual representations and stabilizes visual grounding.

**Effect on hallucination.** To examine whether the improvements in visual grounding translate into reduced hallucination, rather than mere suppression of generation, we evaluate our method on two hallucination-focused benchmarks: CHAIR (Rohrbach et al., 2018) on the COCO validation split (500 images), and AMBER-G (Wang et al., 2023) on its full discriminative split (1,004 images). As shown in Table 5, our method consistently reduces both CHAIR$_s$ and CHAIR$_i$

*Table 5.* **Hallucination evaluation on CHAIR and AMBER-G.** Lower is better for CHAIR$_s$, CHAIR$_i$, AMBER CHAIR, and HAL; higher is better for Cover.

| Method | CHAIR (COCO) | | AMBER-G | | |
|---|---|---|---|---|---|
| | CHAIR$_s \downarrow$ | CHAIR$_i \downarrow$ | CHAIR$\downarrow$ | Cover$\uparrow$ | HAL$\downarrow$ |
| LLaDA-V | 29.4 | 9.5 | 7.2 | 61.6 | 41.1 |
| + Ours | **27.0** | **8.3** | **6.9** | 61.6 | **40.6** |

on LLaDA-V, while AMBER-G shows lower CHAIR and HAL with coverage preserved. This indicates that the gains stem from improved visual grounding rather than degenerate or overly conservative generation. These results are consistent with prior findings that strengthening attention to visual tokens reduces hallucination (Huang et al., 2024; Arif et al., 2025; Favero et al., 2024); Figure 7(b) confirms that our method increases attention from generation tokens toward distant visual tokens.

**Effect on visual spatial structure.** A natural concern is that frequency rescaling may distort the 2D visual spatial structure when applied uniformly across token segments. We verify this is not the case: applying MRS only to visual tokens already improves grounding on both backbones (e.g., $36.9 \rightarrow 43.8$ on RefCOCOg for LaViDa), indicating that monotonic low-frequency amplification preserves 2D positional reasoning rather than damaging it. The full segment-wise comparison (visual-only / textual-only / generation-only / all) is provided in Section I.3.

# 7. Conclusion

In this work, we analyze fundamental challenges of LD-VLMs, including repetitive generation and degraded visual grounding. We identify mask representation drift and positional attention collapse as the primary causes of these issues. To mitigate them, we propose two training-free inference-time techniques: Mask Prior Suppression and Monotonic RoPE Scaling. Our methods integrate seamlessly into existing LDVLMs and consistently improve visual grounding and long-form generation across diverse benchmarks, without sacrificing general reasoning performance. As an inference-time approach, the effectiveness of our method depends on the quality of the underlying pre-trained representations. Our experiments are conducted under the standard full-suffix iterative unmasking setup adopted by current LDVLMs; preliminary results in Section E suggest that our methods remain effective when suffix redundancy is partially reduced (e.g., DPad-style decoding), but extending the analysis to alternative decoding paradigms such as semi-autoregressive or block-wise diffusion variants requires additional multimodal adaptation and is left to future work. We hope this work encourages further research toward more stable and interpretable LDVLMs.

## Impact Statement

This work focuses on improving the stability and reliability of LDVLMs through analysis and inference-time techniques. However, as with other advances in multimodal generation, these techniques could also be applied to systems that generate misleading or harmful content. We emphasize that our methods do not introduce new capabilities beyond those of existing models, and that responsible deployment should follow established safety and ethical guidelines.

## Acknowledgements

This work was supported in part by the IITP RS-2024-00457882 (AI Research Hub Project), IITP 2020-II201361, NRF RS-2024-00345806, NRF RS-2023-002620, NRF-2024S1A5C3A03046579, and RQT-25-120390. Affiliations: Department of Artificial Intelligence (S.H, C.Y, S.J.H).

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

# A. Benchmark Details

## A.1. General

**MME.** (Fu et al., 2025) The MME benchmark is designed to evaluate both perceptual and cognitive abilities of multimodal models. It consists of 14 subtasks, each formulated as a binary (yes/no) question. The perceptual evaluation includes tasks such as object existence and object counting, while the cognitive evaluation covers tasks including commonsense reasoning and text translation.

**MMBench.** (Liu et al., 2024c) The MMBench evaluates the multidimensional capabilities of multimodal models using 3,217 multiple-choice questions organized into a three-level hierarchy. It covers perception and reasoning at the first level, six specialized capability groups (e.g., attribute reasoning and coarse perception) at the second level, and 20 fine-grained abilities (e.g., physical properties and image style) at the final level.

**MMMU.** (Yue et al., 2024) The MMMU is a benchmark designed to evaluate advanced academic-level knowledge and reasoning capabilities of multimodal models. It consists of six subject categories: Art & Design, Business, Science, Health & Medicine, Humanities & Social Science, and Technology & Engineering. The benchmark includes diverse visual modalities beyond natural images, such as charts, musical scores, and chemical structure diagrams. Due to its inclusion of complex and high-level tasks, MMMU is regarded as one of the most challenging benchmarks for MLLMs.

## A.2. Visual Grounding

**RefCOCOg.** (Kazemzadeh et al., 2014) RefCOCOg is a benchmark designed to evaluate a model's ability to comprehend referring expressions and segment specific objects. It is composed of samples that involve long descriptions and demand complex reasoning, presenting a high level of difficulty.

**Ferret.** (You et al., 2024) Ferret evaluates referring and grounding capabilities, which are divided into description, referring reasoning, and grounding in conversational settings. The dataset is constructed by sampling 40 images from the COCO validation set (Lin et al., 2014) and generating corresponding reference responses using GPT-4.

**GQA.** (Hudson & Manning, 2019) The GQA is a benchmark for real-world visual question answering (VQA). In addition to images and questions, it incorporates scene graphs that encode object attributes, spatial positions, and relationships between objects. The dataset balances answer distributions to reduce language priors. Through this benchmark, models can be evaluated on their ability to understand complex relational structures within images.

## A.3. Long-form Generation

**LLaVA Bench (In-the-Wild).** (Liu et al., 2023) The LLaVA-Bench (In-the-Wild) benchmark evaluates a model's creativity and comprehension through open-ended responses, focusing on challenging tasks and generalization to unseen domains. The benchmark evaluates performance on challenging tasks and generalization to unseen domains. It consists of 60 questions associated with 24 images covering diverse real-world scenes.

**DetailCaps.** (Dong et al., 2024) DetailCaps is designed to evaluate the performance of models in image captioning tasks. Its evaluation datasets are constructed through rigorous annotation by human experts and advanced MLLMs such as GPT-4V. It is useful in mitigating hallucinations and enabling analysis of a more diverse range of visual elements.

**MIA.** (Qian et al., 2025) The MIA benchmark evaluates whether a model correctly follows the instructions given. It consists of 400 image–prompt pairs, where each sample includes specific constraints, such as sentence length and output format. GPT-4o is used as an automatic judge to assess not only answer correctness but also instruction adherence.

## A.4. Hallucination

**CHAIR.** (Rohrbach et al., 2018) CHAIR (Caption Hallucination Assessment with Image Relevance) measures object hallucination in image captioning by comparing the set of objects mentioned in a generated caption against the ground-truth object set defined by COCO annotations. We report two standard variants: $CHAIR_s$ (sentence-level: fraction of sentences containing at least one hallucinated object) and $CHAIR_i$ (instance-level: fraction of mentioned objects that are hallucinated).

**AMBER-G.** (Wang et al., 2023) AMBER is an LLM-free hallucination benchmark covering both generative (AMBER-G)

*Table 6.* **Evaluation Setup.** Evaluation splits, inference steps, and generation length $L$ for each benchmark.

| Dataset | Split | Steps | $L$ | Dataset | Split | Steps | $L$ | Dataset | Split | Steps | $L$ |
|---------|-------|-------|-----|---------|-------|-------|-----|---------|-------|-------|-----|
| MME | test | 2 | 2 | Ferret | test | 48 | 96 | DetailCaps | test | 128 | 256 |
| MMBench | en-dev | 2 | 2 | GQA | lite | 2 | 2 | LLaVA-Bench | train | 64 | 128 |
| MMMU | val | 2 | 2 | RefCOCOg | val | 4 | 8 | MIA | test | 32 | 64 |

*Table 7.* **Comparison of training data scale and model size across LDVLMs and baselines.** Dashes (-) indicate results not reported. L-OV; LLaVA-OneVision-7B; Qwen2.5: Qwen2.5-VL-7B; Intern3: Intern-VL3-8B

| | LLaDA-V | LaViDa | LLaVA-1.6 | LLaVA-OV | Qwen2.5 | Intern3 |
|---|---------|--------|-----------|----------|---------|---------|
| # Params | 8B | 8B | 7B | 7B | 7B | 8B |
| # Images (Pretrain) | 0.6M | 0.6M | 0.6M | 0.6M | >7M | – |
| # Images (SFT) | 15.9M | 1.0M | 0.7M | 7.2M | ∼2M | 21.7M |

and discriminative settings. We use the generative split, which evaluates open-ended captions against a curated object inventory and reports CHAIR (object hallucination), Cover (object recall), and HAL (hallucination rate over generated samples).

## B. Evaluation Setup.

We conduct all evaluations on LLaDA-V and LaViDa using the LMMs-Eval framework (Zhang et al., 2024). For all benchmarks, we adopt the default prompts provided by the framework. The evaluation split and the generation length $L$ for each dataset are reported in Table 6. For autoregressive baselines, including LLaVA-One-Vision-7B, Qwen2.5-VL-7B, InternVL3-8B, and LLaVA-1.6, we use the default evaluation setups provided by the same framework. To ensure a rigorous and fair comparison, we evaluate models under identical random seeds whenever reported results are unavailable. Notably, for LaViDa, we conduct a re-evaluation using the same evaluation setups as LLaDA-V, thereby facilitating a strictly controlled comparison.

### B.1. Details of Comparison Models

As shown in Table 7, we summarize the parameter counts and training data scale of the compared models.

### B.2. Inference Details.

We conduct our experiments on a system equipped with one NVIDIA A100-SXM4-80GB GPU and two NVIDIA RTX A6000 GPUs. All experiments are conducted under the recommended environments provided by the respective models to ensure fair evaluation.

## C. Justification of the Mask Prior Direction

We provide additional empirical justification for the choice of the vocabulary mean as our prior direction. Our claim throughout the paper is empirical and limited in scope: we do not assert that the vocabulary mean fully characterizes the nonlinear dynamics of mask-token collapse, only that it captures a dominant, low-dimensional bias direction in the final hidden-state space that is consistently aligned with contextualized mask states. Motivated by the Linear Representation Hypothesis (Park et al., 2024), which suggests that meaningful concepts in deep models can emerge as approximately linear directions in representation space, we adopt this simple and parameter-free proxy.

**Comparison against alternative prior directions.** We compare four candidate directions in the final hidden-state space of LLaDA-V: the vocabulary-mean direction $e_{\text{uniform}}$ (ours), a frequency-weighted lexical prior $e_{\text{freq}}$ estimated from COCO train2014 caption frequencies, the contextualized mask-token direction $e_{\text{mask}}$ averaged over generation, and a random direction $e_{\text{random}}$. Table 8 reports cosine similarities between these candidate directions. The vocabulary-mean direction is highly aligned with the frequency-weighted lexical prior (0.897), indicating that they capture a very similar dominant lexical direction in practice. Both $e_{\text{uniform}}$ and $e_{\text{freq}}$ are substantially more aligned with $e_{\text{mask}}$ (0.883, 0.735) than a random direction (0.427), supporting the use of the vocabulary mean as a stable proxy for a shared collapse direction.

*Table 8.* **Cosine similarity between candidate prior directions in the final hidden-state space (LLaDA-V).** $e_{\text{uniform}}$: vocabulary mean (ours); $e_{\text{freq}}$: frequency-weighted lexical prior; $e_{\text{mask}}$: contextualized mask-token direction averaged over generation; $e_{\text{random}}$: random direction.

| Direction A | Direction B | Cosine Similarity |
| --- | --- | --- |
| $e_{\text{uniform}}$ | $e_{\text{freq}}$ | $+0.897$ |
| $e_{\text{uniform}}$ | $e_{\text{mask}}$ | $+0.883$ |
| $e_{\text{freq}}$ | $e_{\text{mask}}$ | $+0.735$ |
| $e_{\text{random}}$ | $e_{\text{mask}}$ | $+0.427$ |

*Table 9.* **Stability of context-free prior alignment across generation steps.** Cosine similarity between the vocabulary mean and contextualized mask-token hidden states for varying generation steps $T$ (LLaDA-V).

| $T$ | 32 | 16 | 8 | 4 |
| --- | --- | --- | --- | --- |
| Cosine similarity | 0.255 | 0.256 | 0.268 | 0.283 |

**Stability across generation steps.** We further verify that the alignment between the context-free prior direction and contextualized mask-token states is stable across decoding configurations. Table 9 reports the cosine similarity between the vocabulary mean and the contextualized mask-token hidden states for $T \in \{32, 16, 8, 4\}$ generation steps. The alignment is consistently strong and varies only mildly with $T$, while a random direction yields substantially lower alignment. This stability supports the use of a context-free prior estimate throughout iterative unmasking.

**Discussion.** Together with Table 4 in the main paper, which compares the vocabulary mean against top-$k$ token-based priors and random directions in terms of downstream task performance, these results support our design choice on three grounds: (i) the vocabulary mean and frequency-weighted lexical prior are nearly equivalent dominant directions; (ii) both align substantially more with mask-token hidden states than random directions; and (iii) this alignment is stable across decoding configurations. We therefore use the vocabulary mean as a simple, principled proxy for the shared bias component, rather than as a fundamentally different prior from those propagated through the network.

## D. Limitations of Existing Rotary Scaling Methods in LDVLMs

Prior work has explored various rotary scaling strategies to extend the context length of autoregressive language models. NTK-aware RoPE scaling (LocalLLaMA Community, 2023; Liu et al., 2024b) modifies the base frequency of rotary embeddings based on Neural Tangent Kernel analysis, stabilizing attention patterns when extending the context window beyond the training length. Similarly, YaRN (Peng et al., 2024) introduces a piecewise frequency rescaling scheme that preserves high-frequency components while smoothly extrapolating to longer sequences. Subsequent methods further refine rotary scaling to enhance extrapolation stability and efficiency in LLMs (Ding et al., 2024). While these approaches are effective for extending context length under causal decoding, they are primarily designed for autoregressive language modeling. In contrast, LDVLMs employ bidirectional attention and iterative parallel unmasking, resulting in fundamentally different attention dynamics. Consequently, existing rotary scaling methods do not directly address the attention imbalance and visual grounding challenges that arise in LDVLMs. The empirical comparison with NTK and YaRN is reported in Table 3 of the main paper, where these baselines often degrade visual grounding while our monotonic low-frequency amplification yields consistent gains across both backbones.

## E. Comparison with Decoding-Level Baselines

We further compare our method against a decoding-level intervention that targets suffix-mask redundancy directly during inference. We adopt a DPad-style suffix-dropping baseline, which is training-free and modifies suffix attention at the input level, making it directly applicable to existing LDVLM checkpoints. This comparison is informative because DPad and our method address related but distinct mechanisms: DPad reduces redundant suffix-mask influence at the input level, whereas our Monotonic RoPE Scaling strengthens long-range access to informative visual tokens at the representation level.

Table 10 shows that our method consistently outperforms DPad alone on both backbones across all three benchmarks. On LaViDa, DPad and our method are largely complementary: combining them yields the best overall results. On LLaDA-V, the combination is mixed: DPad's input-level masking may partially remove positional context that our Monotonic RoPE

*Table 10.* **Comparison with a DPad-style decoding-level baseline.** Our method is complementary to DPad on LaViDa, where the combination yields the best results across all three benchmarks.

| Method | LLaDA-V | | | LaViDa | | |
|---|---|---|---|---|---|---|
| | RefCOCOg | Ferret | DetailCaps | RefCOCOg | Ferret | DetailCaps |
| Baseline | 64.8 | 60.4 | 59.8 | 36.9 | 25.9 | 8.3 |
| + DPad | 64.7 | 60.3 | 60.2 | 39.6 | 30.8 | 7.9 |
| + Ours | **65.0** | **62.9** | **63.6** | 44.0 | 35.7 | 56.1 |
| + DPad + Ours | **65.0** | 61.7 | 60.5 | **44.4** | **36.1** | **56.6** |

Scaling exploits, which limits additivity in this setting.

**On D2F.** D2F (diffusion-to-feed-forward) replaces the decoding paradigm with a block-wise autoregressive–diffusion hybrid, realized through asymmetric distillation from a pretrained DLLM, rather than as a training-free inference-time modification. Moreover, it is introduced for text-only DLLMs. Adapting D2F to existing LDVLM checkpoints would therefore require additional multimodal adaptation or training, which falls outside the scope of our training-free analysis. We view it as a complementary direction and an interesting target for future work.

## F. Baselines

**LLaDA-V.** (You et al., 2025) LLaDA-V adopts the LDVLM framework, building on the LLaDA (Nie et al., 2025b) architecture. The model comprises a SigLIP2-SO400M-Patch14-384 vision encoder (Tschannen et al., 2025) and an LLaDA-8B-Instruct language model. During training, response tokens in a multi-turn dialogue are randomly masked and predicted via bidirectional attention. LLaDA-V follows a three-stage training paradigm. In the first stage, the model is trained on the LLaVA-Pretrain (Liu et al., 2023) dataset to align visual and linguistic representations. In the second stage, visual instruction tuning is performed using the MAmmoTH-VL (Guo et al., 2025) dataset to enhance multimodal understanding. In the final stage, multimodal reasoning capabilities are further strengthened using the VisualWebInstruct (Jia et al., 2025) dataset, which contains approximately 900K question–answer pairs involving complex visual reasoning tasks. Overall, LLaDA-V demonstrates performance comparable to autoregressive VLMs, highlighting its potential as an alternative modeling paradigm.

**LaViDa.** (Li et al., 2025b) LaViDa is an LDVLM proposed to address limitations of autoregressive MLLMs, particularly their slow inference speed and limited controllability during generation. LaViDa employs a SigLIP-400M vision encoder (Zhai et al., 2023) and an LLM backbone based on LLaDA-8B or Dream-7B (Ye et al., 2025). In our experiments, we use LaViDa-L only, as it shares the same language backbone as LLaDA-V, enabling a fair comparison. LaViDa introduces a complementary masking strategy during training. Instead of learning from a single masked version of a response, two complementary masked variants are constructed for each sample such that the masked regions do not overlap. This design ensures that all tokens contribute to training across iterations, alleviating efficiency and performance degradation caused by sparse masking. LaViDa follows a two-stage training paradigm. In the first stage, the model is pretrained by updating only the MLP projector to align visual embeddings with the LLM's word embedding space, using the LCS-558K dataset of 558K image–text pairs. In the second stage, the entire model, including the vision encoder and the LLM, is jointly finetuned on approximately one million samples collected from multiple datasets such as COCO and ALLaVA-VFLAN (Chen et al., 2024). During inference, LaViDa further proposes Prefix-DLM, an inference optimization technique that caches and reuses the key–value states of prefix tokens, including visual tokens and prompt tokens. This caching mechanism reduces redundant computation and improves inference speed. In addition, LaViDa replaces the commonly used linear unmasking schedule with a shifting schedule, leading to further performance improvements.

## G. Additional Experiments on Unified Models

We further apply our method to a unified model architecture. Both models are evaluated using VLMEvalKit (Duan et al., 2024) under a consistent evaluation protocol. For MMaDA (Yang et al., 2025), we set $\lambda = 0.1$, $\beta = 0.4$, $k = 3$, $\eta = 8.0$, and $\tau_0 = 0.6$. For Lumina-DiMOO (Xin et al., 2025), we set $\lambda = 0.1$, $\beta = 0.4$, $k = 3$, $\eta = 12.0$, and $\tau_0 = 0.6$. In both cases, our method consistently outperforms the corresponding baselines, as shown in Table 11. These results demonstrate that our approach generalizes beyond LDVLMs and can be effectively applied to unified vision–language architectures.

*Table 11.* **Quantitative Results on Unified Models** Lumina refers to Lumina-DiMOO.

|  | MME-P | MMMU | MMB |
|---|---|---|---|
| MMaDA (reported) | 1410.7 | 30.2 | 68.5 |
| MMaDA (reproduced) | 1070.6 | 31.1 | 44.5 |
| +Ours | 1117.2 | 32.7 | 44.5 |
| Lumina (reported) | 1534.2 | 58.6 | 84.5 |
| Lumina (reproduced) | 1543.6 | 59.3 | 84.9 |
| +Ours | 1553.1 | 62.0 | 85.0 |

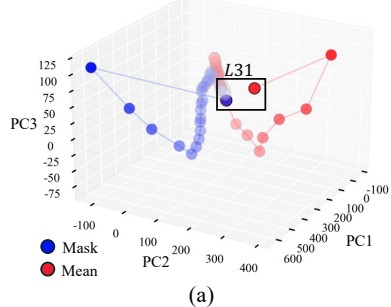
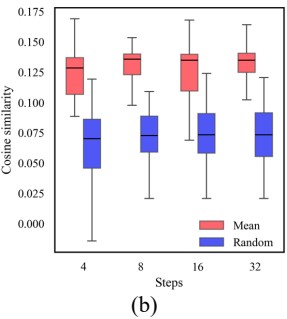

(a)           (b)

*Figure 8.* **Visualization of mask prior drift on LaViDa.** (a) 3D PCA trajectories of hidden states for the vocabulary mean embedding and the uncontextualized mask token, which converge to a similar region at the final layer ($L31$). (b) Cosine similarity between contextualized mask token embeddings and the vocabulary mean, showing consistently stronger alignment than random embeddings, especially with fewer generation steps.

# H. Additional Analysis and results on LaViDa and LLaDA-V

## H.1. Analysis and Results on LaViDa

In this section, we provide additional visual analyses of LaViDa to further support the observations presented in the main paper. Specifically, we extend the analysis of mask token drift and positional attention collapse to LaViDa architecture. As shown in Figs. 8 and 9, the qualitative trends remain highly consistent with our previous findings, confirming that these phenomena are inherent characteristics of LDVLMs rather than artifacts of a specific model instance. Moreover, as illustrated in Fig. 10, LaViDa exhibits the same behavior observed in Section 6.3, where the mask prior is effectively mitigated, and attention to visual tokens is preserved even at long relative distances.

## H.2. Additional Results on LLaDA-V

As shown in Figure 11, both distinct-$n$ and repetition ratios exhibit step-dependent improvements over the baseline, with more pronounced gains observed at intermediate generation steps.

## H.3. Generation Step Analysis on DetailCaps

We further analyze the relationship between generation steps and DetailCaps performance on LaViDa. As shown in Figure 12(a), the uncontextualized mask token $\mathcal{M}$ exhibits a strong bias toward the |eot| token, which attains the highest logit among all vocabulary sets. This indicates that initialized generation tokens as $\mathcal{M}$ inherently leads to early termination. This bias is reflected in the quantitative results on the DetailCaps benchmark. As illustrated in Figure 12(b), increasing the number of generation steps does not consistently improve caption quality. Instead of following a conventional speed–quality trade-off, the CAPTURE score peaks at 16 steps and subsequently decreases as the number of steps increases. This behavior contrasts with the expectation that longer decoding should yield more detailed descriptions. The underlying reason is closely related to mask prior drift. As shown in Figure 12(c), when a large number of steps (e.g., 64, 128) is used, the |eot| token is repeatedly generated across steps, causing captions to terminate prematurely. In contrast, with a moderate number of steps, more tokens are generated in parallel, reducing the dominance of the |eot| prior and mitigating early stopping. This explains why LaViDa exhibits notably low DetailCaps performance in Table 2, and highlights the strong connection between mask prior drift and degraded long-form caption quality.

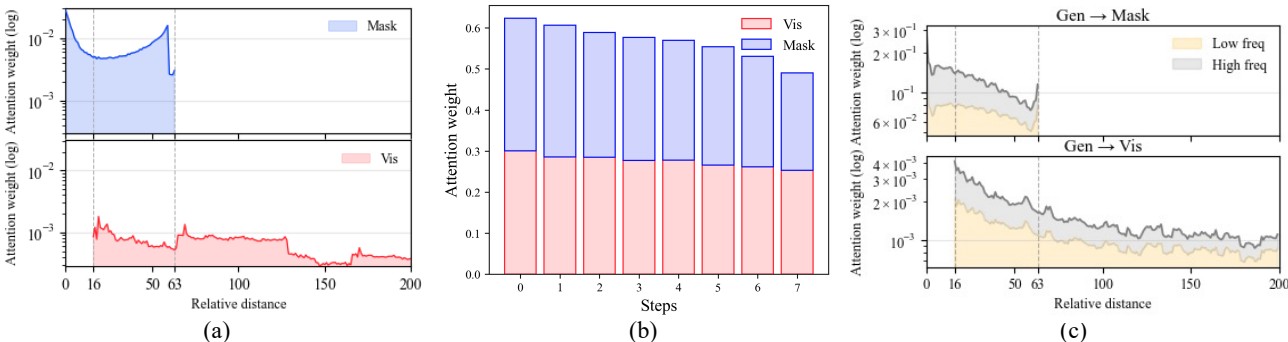

*Figure 9.* **Visualization of Positional Attention Collapse on LaViDa** (a) Mean attention weight across relative distance (log scale), showing stronger attention to mask tokens than visual tokens at similar distances and a monotonic decay for visual tokens. (b) Sum of attention to visual and mask tokens per generation token across generation steps, revealing a persistent allocation of comparable attention weights to mask tokens despite their lack of semantic content. (c) Frequency decomposition of attention over relative distance, with high-frequency dominance at short ranges and low-frequency dominance at long ranges, yet weak overall attention.

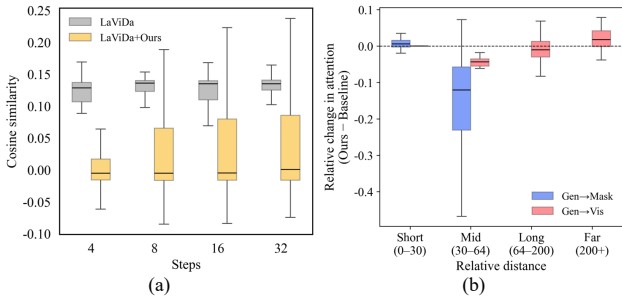

*Figure 10.* **Visualization of result analysis on LaViDa.** (a) Box plot of cosine similarity between contextualized mask tokens and the vocabulary mean, showing consistent reduction across generation steps. (b) Relative change in attention with respect to relative distance, where attention to distant visual tokens increases compared to the baseline, while attention to mask tokens is preserved or reduced.

# I. Additional Experimental Results

### I.1. Reference-Based Captioning Metrics on RefCOCOg

We further report standard reference-based captioning metrics on RefCOCOg, which provides 7,573 images with human-annotated ground-truth references. Table 12 shows that our method matches or improves upon the baselines across CIDEr, BLEU-4, METEOR, and RefCLIPScore. For DetailCaps and LLaVA-Bench, standard reference-based metrics are less directly applicable: DetailCaps does not provide canonical reference captions, and LLaVA-Bench uses GPT-4 judge scores as its evaluation protocol. We therefore report task-level scores in the main tables for these benchmarks.

### I.2. Layer-wise Application of Mask Prior Suppression

Our default configuration applies Mask Prior Suppression (MPS) only at the final transformer layer. We ablate this design by progressively expanding MPS to deeper subsets of layers. Table 13 shows that applying MPS at the final layer alone provides the most stable trade-off across benchmarks and is consistently strongest on long-form generation (LLaVA-Bench, DetailCaps). Suppressing the prior at earlier layers can interfere with intermediate representations that still encode useful contextual information, particularly hurting long-form generation on LaViDa. The reported *prior* column denotes the average ratio of suppressed prior energy across the configured layers; with a fixed total budget, applying MPS to more layers necessarily distributes per-layer suppression more thinly.

### I.3. Segment-wise Application of Monotonic RoPE Scaling

A natural concern is that frequency rescaling might distort 2D visual spatial structure when applied uniformly across visual, textual, and generation tokens. Although Monotonic RoPE Scaling rescales RoPE frequency components without changing token indices, patch layout, or sequence order, we provide a direct empirical check by restricting the rescaling to specific

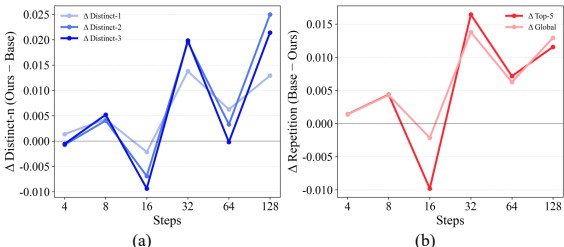

(a)                          (b)

*Figure 11.* **Relative performance changes on DetailCaps across generation steps using LLaDA-V.** Dashed lines: LLaDA-V, solid lines: Ours. (a) $\Delta$Distinct-$n$ (Ours – Base) shows consistent gains, with larger improvements at moderate to larger generation steps. (b) $\Delta$Repetition ratio (Base – Ours) remains positive across most steps, indicating reduced repetition, with the strongest reductions observed at intermediate steps.

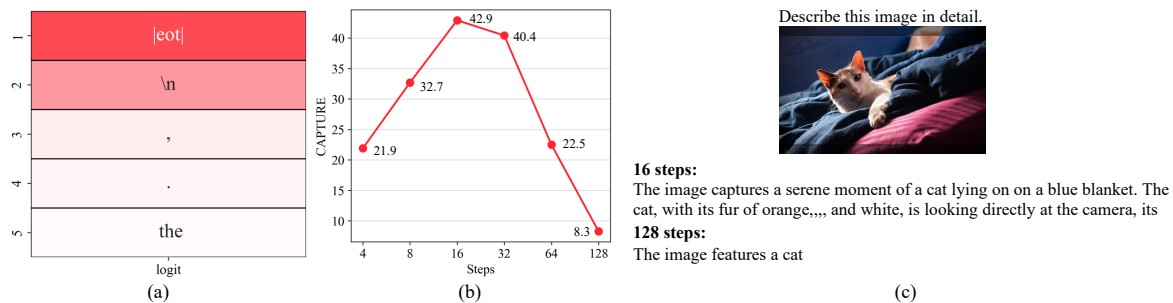

(a)                          (b)                          (c)

*Figure 12.* **Generation step analysis and DetailCaps performance on LaViDa.** (a) Top-5 logits of the uncontextualized mask token $\mathcal{M}$, where the `|eot|` token consistently receives the highest logit. (b) CAPTURE scores on the DetailCaps benchmark as a function of generation steps. Contrary to a standard speed–quality trade-off, performance peaks at 16 steps and degrades with additional steps. (c) Qualitative examples showing early termination induced by mask token prior at larger step counts (e.g., 32, 64, 128), where frequent generation of `|eot|` results in prematurely truncated captions.

token segments. Table 14 reports the results when MRS is applied only to visual tokens, only to textual prompt tokens, only to generation tokens, or to all tokens (default).

A few observations follow. First, applying MRS to visual tokens alone already improves visual grounding on both backbones, indicating that the rescaling does not degrade 2D visual spatial structure. Second, no single segment consistently dominates: on LLaDA-V, applying MRS to all tokens is best on DetailCaps; on LaViDa, applying MRS to either visual or textual tokens alone is best on DetailCaps but very close to the all-tokens setting on the remaining benchmarks. Overall, the uniform all-tokens setting provides the strongest or near-strongest performance across benchmarks without task-specific tuning, which we adopt as the default.

### I.4. Hyperparameter Sensitivity

We analyze the hyperparameter sensitivity of our method to assess its robustness. We evaluate the impact of five hyperparameters: $\lambda$ for prior suppression strength, $\beta$ for RoPE scaling magnitude, $k$ for prior subspace dimensionality, $\eta$ for the slope of monotonic RoPE scaling, and $\tau_0$ for the center of the frequency-wise scaling setting. Table 15 and Table 16 summarize the results across multiple benchmarks. Overall, we observe that performance varies smoothly with respect to $\lambda$ and $\beta$, indicating low sensitivity over a broad range of values. In contrast, the choice of $k$ exhibits a clear trade-off between expressiveness and stability: very small values limit the capacity of prior suppression, while excessively large values lead to unstable behavior, especially in long-form generation. Moderate values such as $k = 3$ consistently provide a favorable balance across benchmarks. We further find that the method is relatively robust to the RoPE scaling parameters $\eta$ and $\tau_0$. Varying $\eta$ mainly affects the sharpness of frequency-wise reweighting, with moderate slopes yielding stable performance, while extreme values provide limited additional benefits. Similarly, adjusting $\tau_0$ results in only minor performance fluctuations, suggesting that the method does not rely on precise tuning of the frequency center. These observations indicate that the proposed monotonic RoPE scaling is stable across a wide range of configurations.

*Table 12.* **Reference-based captioning metrics on RefCOCOg.** Higher is better for all metrics.

| Method | CIDEr | BLEU-4 | METEOR | RefCLIPScore |
|---|---|---|---|---|
| LLaDA-V | 64.8 | 8.2 | 14.6 | 72.8 |
| + Ours | **65.0** | **8.4** | 14.6 | 72.8 |
| LaViDa | 36.9 | 1.5 | 8.4 | 68.1 |
| + Ours | **44.0** | **4.9** | **11.3** | **69.4** |

*Table 13.* **Layer-wise application of Mask Prior Suppression.** *Final-only* denotes our default setting.

| Model | Configuration | Prior | RefCOCOg | LLaVA-Bench | DetailCaps |
|---|---|---|---|---|---|
| LLaDA-V | Baseline | – | 64.8 | 61.3 | 59.8 |
| | All layers (×32) | 0.003 | 61.3 | 63.5 | 60.2 |
| | Last 8 layers | 0.013 | **65.1** | 61.1 | 59.8 |
| | Last 3 layers | 0.033 | 64.8 | 61.1 | 60.3 |
| | Final-only (Ours) | 0.100 | 65.0 | **64.1** | **63.6** |
| LaViDa | Baseline | – | 36.9 | 39.5 | 8.3 |
| | All layers (×32) | 0.009 | 40.0 | 37.4 | 6.7 |
| | Last 8 layers | 0.038 | 39.6 | 36.7 | 8.1 |
| | Last 3 layers | 0.100 | 39.7 | 35.8 | 8.1 |
| | Final-only (Ours) | 0.300 | **44.0** | **46.5** | **56.1** |

## I.5. Inference latency

We evaluate the inference latency of our method under varying numbers of generation steps. As shown in Table 17, our approach introduces a small and consistent latency overhead compared to the base models. The additional latency scales proportionally with the number of generation steps, indicating that the proposed modifications preserve the efficiency of the underlying decoding process.

## I.6. RoPE Scheduling

We study the effect of different RoPE scheduling functions on visual grounding and long-form generation performance. Table 18 compares sigmoid, cosine, exponential, linear, and power schedules under the same setting. Across both LLaDA-V and LaViDa, we observe that the choice of scheduling function has a more pronounced impact on visual grounding than on long-form generation. In particular, the sigmoid schedule consistently yields strong performance on visual grounding benchmarks such as RefCOCOg and Ferret. Compared to alternative schedules, sigmoid provides stable improvements without introducing performance degradation across tasks. We attribute this behavior to the smooth and monotonic nature of the sigmoid schedule, which gradually modulates low-frequency positional components while preserving high-frequency information. This property helps maintain long-range visual–textual alignment, which is critical for visual grounding tasks. In contrast, schedules with more abrupt or aggressive scaling, such as exponential or power functions, can distort positional biases and lead to less stable grounding performance. Overall, these results suggest that smooth, monotonic scheduling functions are better suited for controlling RoPE-induced locality bias in multimodal grounding scenarios.

## J. Qualitative Results

Additional qualitative results on visual grounding and long-form generation for LLaDA-V are provided in Figures 13 to 16. Additional qualitative results for LaViDa are provided in Figures 17 to 20.

## K. Limitations and Future Work

**Limitations.** While our method consistently improves visual grounding and long-form generation without additional training, it is primarily designed for inference-time intervention and does not modify the underlying model parameters. As a result, the effectiveness of our approach depends on the quality of the pretrained representations and may be limited when applied to models with weaker multimodal alignment or different positional encoding schemes. In addition, our analysis focuses on attention dynamics and hidden-state behavior in masked diffusion-based vision-language models, and the generalization of these findings to other generation paradigms remains to be fully explored. We further note that our

*Table 14.* **Segment-wise application of Monotonic RoPE Scaling.** "vis only", "txt only", and "gen only" restrict MRS to visual, textual, or generation tokens; "all" is the default.

| Backbone | Variant | RefCOCOg | Ferret | DetailCaps |
|---|---|---|---|---|
| LLaDA-V | Baseline | 64.8 | 60.4 | 59.8 |
| | vis only | **65.2** | 62.5 | 59.6 |
| | txt only | 65.0 | **63.2** | 60.2 |
| | gen only | 64.9 | 61.8 | 60.1 |
| | all (Ours) | 65.0 | 62.9 | **63.6** |
| LaViDa | Baseline | 36.9 | 25.9 | 8.3 |
| | vis only | 43.8 | 35.9 | **56.4** |
| | txt only | 43.8 | **36.1** | **56.4** |
| | gen only | 43.8 | 35.8 | **56.4** |
| | all (Ours) | **44.0** | 35.7 | 56.1 |

*Table 15.* **Hyperparameter sensitivity analysis on LLaDA-V.** The highlighted row denotes the default configuration used in all main experiments.

| Hyperparameter | | | | | Visual Grounding | | Long-form Generation | |
|---|---|---|---|---|---|---|---|---|
| $\lambda$ | $\beta$ | $k$ | $\eta$ | $\tau_0$ | RefCOCOg | Ferret | LLaVA-Bench | DetailCaps |
| 0.2 | 0.1 | 3 | 8 | 0.6 | 64.8 | 62.4 | 61.9 | 60.0 |
| 0.3 | 0.1 | 3 | 8 | 0.6 | 64.2 | 61.3 | 64.7 | 60.4 |
| 0.1 | 0.2 | 3 | 8 | 0.6 | 65.3 | 61.8 | 59.8 | 66.4 |
| 0.1 | 0.3 | 3 | 8 | 0.6 | 65.2 | 61.7 | 63.9 | 59.7 |
| 0.1 | 0.1 | 1 | 8 | 0.6 | 66.9 | 60.7 | 60.1 | 60.1 |
| 0.1 | 0.1 | 8 | 8 | 0.6 | 64.4 | 60.3 | 61.8 | 60.0 |
| 0.1 | 0.1 | 16 | 8 | 0.6 | 64.5 | 61.7 | 62.5 | 60.1 |
| 0.1 | 0.1 | 3 | 4 | 0.6 | 64.8 | 61.1 | 60.5 | 59.5 |
| 0.1 | 0.1 | 3 | 10 | 0.6 | 64.9 | 61.1 | 63.7 | 60.0 |
| 0.1 | 0.1 | 3 | 8 | 0.4 | 65.3 | 61.2 | 61.9 | 60.0 |
| 0.1 | 0.1 | 3 | 8 | 0.8 | 64.9 | 62.3 | 64.0 | 59.8 |
| 0.1 | 0.1 | 3 | 8 | 0.6 | 65.0 | 62.9 | 64.1 | 63.6 |

experiments are conducted under the standard full-suffix iterative unmasking setup adopted by current LDVLMs. The DPad comparison in Section E indicates that our methods remain effective when suffix redundancy is partially reduced; however, validating our approach in more substantially modified decoding paradigms—such as semi-autoregressive or block-wise diffusion variants like D2F, which require additional multimodal adaptation—remains an important direction for future work.

**Future work.** While this study establishes the efficacy of the proposed mechanisms, several promising directions remain for future investigation. First, we aim to integrate these mechanisms into training-time optimization in order to examine their effects on model convergence and the formation of intrinsic representations. Second, we plan to extend our analysis to alternative positional encoding schemes, including various variants of RoPE and their applications to LDVLMs, to gain a deeper understanding of the generality of our findings. Finally, evaluating the robustness of our approach in complex multimodal settings, such as video-language understanding and long-context multi-turn generation, constitutes an important step toward more general and scalable multimodal systems.

*Table 16.* **Hyperparameter sensitivity analysis on LaViDa.** The highlighted row denotes the default configuration used in all main experiments.

| Hyperparameter | | | | | Visual Grounding | | Long-form Generation | |
|---|---|---|---|---|---|---|---|---|
| $\lambda$ | $\beta$ | $k$ | $\eta$ | $\tau_0$ | RefCOCOg | Ferret | LLaVA-Bench | DetailCaps |
| 0.2 | 0.1 | 3 | 12 | 0.6 | 44.0 | 34.6 | 46.3 | 48.4 |
| 0.4 | 0.1 | 3 | 12 | 0.6 | 44.0 | 43.2 | 43.2 | 57.1 |
| 0.3 | 0.2 | 3 | 12 | 0.6 | 44.0 | 35.0 | 40.8 | 56.1 |
| 0.3 | 0.3 | 3 | 12 | 0.6 | 44.0 | 34.7 | 45.6 | 56.1 |
| 0.3 | 0.1 | 1 | 12 | 0.6 | 39.9 | 29.3 | 41.1 | 8.3 |
| 0.3 | 0.1 | 8 | 12 | 0.6 | 44.7 | 38.4 | 47.8 | 56.8 |
| 0.3 | 0.1 | 16 | 12 | 0.6 | 44.8 | 36.9 | 38.1 | 56.9 |
| 0.3 | 0.1 | 3 | 10 | 0.6 | 44.0 | 35.3 | 44.9 | 56.1 |
| 0.3 | 0.1 | 3 | 14 | 0.6 | 44.0 | 34.7 | 47.3 | 56.1 |
| 0.3 | 0.1 | 3 | 12 | 0.4 | 44.0 | 35.3 | 43.4 | 56.1 |
| 0.3 | 0.1 | 3 | 12 | 0.8 | 44.0 | 35.0 | 44.7 | 56.1 |
| 0.3 | 0.1 | 3 | 12 | 0.6 | 44.0 | 35.7 | 46.5 | 56.1 |

*Table 17.* **Latency under different generation steps.** We measure inference latency by varying the number of generation steps while fixing the generation length to 64.

| Model | Generation Steps | | | |
|---|---|---|---|---|
| | 4 | 8 | 16 | 32 |
| LLaDA-V | 2.31s | 4.56s | 9.07s | 18.10s |
| + Ours | 2.33s | 4.60s | 9.11s | 18.16s |
| LaViDa | 0.49s | 0.92s | 1.79s | 3.52s |
| + Ours | 0.57s | 1.09s | 2.12s | 4.19s |

*Table 18.* **Ablation study on different scheduling functions.** We compare sigmoid, cosine, exponential, linear, and power schedules on visual grounding and long-form generation benchmarks.**Bold**: best, underline: second best.

| Model | Type | Visual Grounding | | Long-form Generation | |
|---|---|---|---|---|---|
| | | RefCOCOg | Ferret | LLaVA-Bench | DetailCaps |
| LLaDA-V | Cosine | **65.1** | **63.4** | 62.7 | 60.6 |
| | Exp | 65.0 | 61.4 | 61.4 | 60.2 |
| | Linear | **65.1** | 61.3 | 62.2 | 60.7 |
| | Power | 64.9 | 63.5 | 61.5 | 60.3 |
| | **Sigmoid** | 65.0 | 62.9 | **64.1** | **63.6** |
| LaViDa | Cosine | 43.8 | 34.3 | 45.9 | **56.6** |
| | Exp | 44.0 | 34.1 | **46.8** | 56.4 |
| | Linear | 43.8 | 32.6 | 48.0 | 48.9 |
| | Power | **44.2** | 34.3 | 43.3 | 55.5 |
| | **Sigmoid** | 44.0 | **35.7** | 46.5 | 56.1 |

Provide a short description for this region.

Provide a short description for this region.

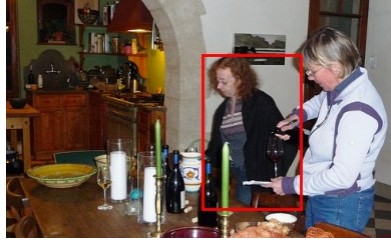

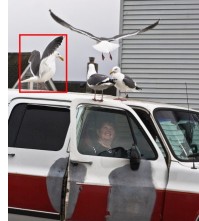

**LLaDA-V:**
Woman woman in glasses pouring wine

**LLaDA-V+Ours:**
A woman pouring a glass wine

**LLaDA-V:**
A woman sitting in a car

**LLaDA-V+Ours:**
Birds on top of car

*Figure 13.* **Qualitative results on RefCOCOg using LLaDA-V.** The red bounding boxes indicate the target regions in the image.

What is the function of the object [0.916, 0.405, 0.934, 0.438]?

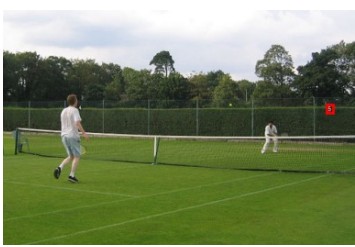

**LLaDA-V:**
The object [0.916, 0.405, 0.934, 0.438] is to be found on a tennis court. It is a metal fence that serves as a boundary for the tennis. The fence is attached to the perimeter of the tennis court and is used to delineate the playing area during the game. The function of this object is to keep the tennis ball within the boundaries of the court

**LLaDA-V+Ours:**
The object [0.916, 0.405, 0.934, 0.438] is to be found as a number on the fence. It is likely used for identification or organization purposes on the tennis court. The number "5" could represent the court number, a player identifier, or some other function related to the game, such as tracking scores, organizing matches, or marking the boundaries of the court

*Figure 14.* **Qualitative results on Ferret using LLaDA-V.** The red bounding boxes indicate the target regions in the image.

What type of fruit is this?

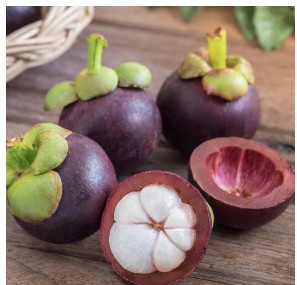

**LLaDA-V:**
The fruit in the image is a mangoes. Mangoes are known for their sweet, juicy flesh and are commonly found in tropical and subtropical regions. They are typically eaten raw but can also be used in desserts, smoothies, and other dishes. The white flesh of mangoes is soft and juicy, with a slightly sweet and tangy flavor. The skin of mangoes is usually dark and can, depending on the variety, be green, red, or purple, indicating the different stages of ripeness. The image shows several whole mangoes and one partially cut open, revealing the white flesh, which is characteristic of mangoes

**LLaDA-V+Ours:**
The fruit in the image is a mangosteens. Mangosteens are a type of tropical fruit that are known for their sweet and and tangy flesh. They are often used in desserts, smoothies, and as a refreshing snack. Mangosteens are native to Southeast Asia and are widely grown in tropical and subtropical regions around the world. They are known for their rich flavor and are often used in various dishes, including desserts, smoothies, and as a refreshing snack. The image shows a bunch of mangosteens, and one of the fruit is cut open, revealing the white flesh, which is characteristic of mangosteens

*Figure 15.* **Qualitative results on LLaVA-Bench using LLaDA-V.**

Describe the image by stating the purpose of the sign and provide the location it references

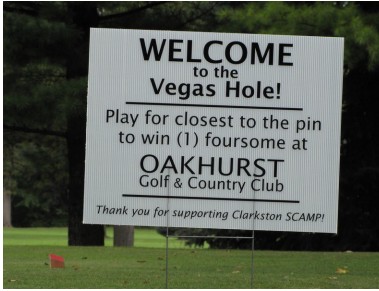

**LLaDA-V:**
The image shows a sign sign that welcomes visitors to the Vegas Hole at Oakhurst Golf & Country Club. The sign encourages players to play for closest to the pin to win a1 four-some. The sign also expresses gratitude for supporting Clarkston SCAMP. The background features a grassy area of the golf course

**LLaDA-V+Ours:**
The image shows a sign that reads "Welcome to the Vegas Hole!" and encourages players to "Play for closest to the pin to win (1)oursome at Oakhurst Golf & Country Club." The sign is intended to attract players to participate in a special golf hole at the Oakhurst Golf & Country Club

*Figure 16.* **Qualitative results on MIA using LLaDA-V.**

Provide a short description for this region.    Provide a short description for this region.

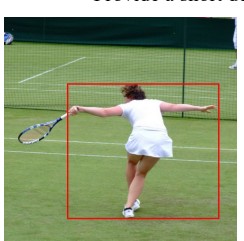
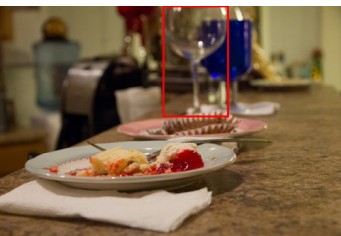

**LaViDa:**
The net of a tennis court.

**LaViDa+Ours:**
A woman holding a tennis racket.

**LaViDa:**
Red plate

**LaViDa+Ours:**
Glass of wine on the table.

*Figure 17.* **Qualitative results on RefCOCOg using LaViDa.** The red bounding boxes indicate the target regions in the image.

What does the region [0.646, 0.869, 0.824, 0.923] likely represent in this image?

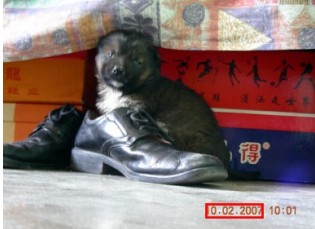

**LaViDa:**
The time a clock.

**LaViDa+Ours:**
The time stamp on the photo.

What are the details of the region [0.279, 0.524, 0.341, 0.570] and how does it relate to the nearby objects?

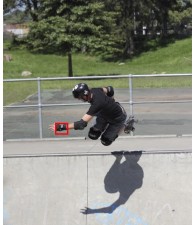

**LaViDa:**
The hand of a person.

**LaViDa+Ours:**
The man is wearing a glove.

*Figure 18.* **Qualitative results on Ferret using LaViDa.** The red bounding boxes indicate the target regions in the image.

Show the detailed recipe for this dish.

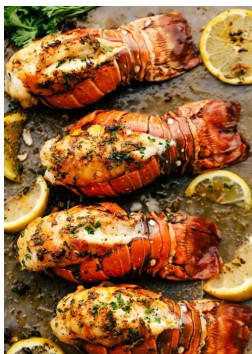

**LaViDa:**
None

**LaViDa+Ours:**
The image provided a visual representation of the dish, but without textual information or additional context, it is not possible to provide a detailed recipe. Typically, a recipe for a dish would include the ingredients, cooking methods, and any specific techniques or steps. In this case, the recipe would likely specify the type of lobster, the amount of seasoning, and the cooking time, as well as any additional ingredients or garnishes. To provide a detailed recipe, one would need to refer to the original source of the dish, such as a cookbook or a menu from a restaurant, which would typically include all the necessary information to recreate the dish.

*Figure 19.* **Qualitative results on LLaVA-Bench using LaViDa.**

Express the feelings that might be elicited by this image using a first-person perspective, specify any author's name visible, while implying a sense of nostalgia.

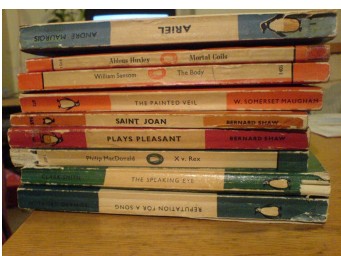

**LaViDa:**
This, this image evokes a sense of nostalgia and the. The, the stack of the,, the, the, the, the, the, the, the, the, the, the, the, the, the, the, the, the, the, the, the, the the.

**LaViDa+Ours:**
This image evokes a sense of nostalgia and the timeless appeal of classic literature. The presence of classics like "The Soul of" by George Bernard Shaw and "The Paintedcible" by William Goldle suggests a collection of works that have been cherished over the years, invoking a sense of the past continuing in the present.

*Figure 20.* **Qualitative results on MIA using LaViDa.**

