# OpenReview forum: "Mitigating Mask Prior Drift and Positional Attention Collapse in Large Diffusion Vision-Language Models"
_ICML.cc/2026/Conference — ICML 2026 regular_

### Official Review · Reviewer_yd9x · 2026-03-08

**Soundness:** 3
**Presentation:** 3
**Significance:** 3
**Originality:** 3
**Overall Recommendation:** 5
**Confidence:** 3

**Summary:**

This paper explores two main challenges faced by LDVLMs in generating long text: repetitive text generation and decreased visual localization capabilities. Through in-depth analysis, the authors point out that these problems stem from "mask prior drift" (the hidden representations of generated tokens gradually shift towards a shared prior) and "positional attention collapse" (local bias of RoPEs causes the model to overemphasize nearby mask tokens while neglecting informational visual tokens at a distance). To address these issues, the paper proposes two inference-stage intervention methods that require no additional training: Mask Prior Suppression (MPS) and Monotonic RoPE Scaling (MRS). Experiments on models such as LLaDA-V and LaViDa demonstrate that this plug-and-play approach effectively improves the quality of long text generation and the accuracy of visual localization.

**Compliance With Llm Reviewing Policy:**

Affirmed.

**Final Justification:**

Thank the authors for the detailed response; I am very satisfied with the reply, it has resolved most of my questions. Therefore, I raised my rating to 5: accept.

The authors' perspective on utilizing PCA for exploration is quite interesting. In fact, [a] similarly utilizes PCA to explore how the visual encoder "captures an important, practically useful component" that impacts downstream tasks. This represents a clever approach to controlling and investigating the internal information flow within the model. I hope the authors will consider incorporating this discussion into the final version of the paper to further enrich its contribution to the community.

[a] Visual Instance-aware Prompt Tuning. ACM MM 25

**Key Questions For Authors:**

Please see the Weaknesses part.

**Limitations:**

Yes

**Strengths And Weaknesses:**

# Strengths:

This paper explores the generation failure modes unique to LDVLMs. The proposed solution operates primarily during the inference phase, requiring no parameter modification or retraining, providing a lightweight and universal intervention strategy. It addresses a key bottleneck in the generation of complex multimodal long texts using diffusion-visual-language models, offering a more stable technical path to replace traditional autoregressive models.

# Weaknesses:

The theoretical foundation of the mask prior definition is somewhat weak: The paper simply approximates the "mask prior" as the arithmetic mean of the word embeddings $\hat{e}$, and uses principal component analysis (PCA) to obtain an orthogonal basis $U$ for linear projection and suppression. From the perspective of information theory and high-dimensional representation learning, the trajectory evolution of mask tokens in deep Transformers is highly nonlinear and involves complex contextual interactions. Simply removing the linear mean principal components in the last layer of the feature space may only deal with the appearance (i.e., the averaging of the word distribution), without addressing the fundamental mechanism of information entropy collapse during parallel decoding.

Potential Distortion of 2D Visual Spatial Structure by Monotonic RoPE Scaling: The proposed monotonic RoPE scaling (MRS) is essentially a simple amplification of the low-frequency components of a one-dimensional sequence based on Sigmoid gates. However, visual tokens have an inherent two-dimensional spatial topology. While amplifying low-frequency components forces the model to focus on distant tokens, directly imposing this frequency-related penalty on a 1D unfolded multimodal sequence may distort the relative spatial relationships between image patches. The paper lacks a theoretical analysis of how this scaling affects the structure of 2D visual representations.

Potential Interference with Learned Diffusion Dynamics during Inference: The core of LDVLMs is learning the conditional distribution $p_\theta(x_0|x_t)$ of the backward diffusion process. Forcibly modifying the hidden state in the last layer during inference to suppress prior features $\tilde{h}_{L}^{e_j}$ artificially disrupts the Markov transition probabilities originally learned by the model. This superficial post-processing intervention, while suppressing repetitive generation on the current evaluation set, may lead to unpredictable artifacts generated in out-of-distribution (OOD) scenarios or more complex inference tasks because the model no longer strictly adheres to its score-matching objective from the training phase.

Insufficient Decoupling of Grounding and Hallucination: The paper claims that the MRS mechanism addresses the decline in grounding by altering attention distribution and demonstrates score improvements on benchmarks such as RefCOCOg and Ferret. However, this fails to answer a core question: Does the model truly enhance its understanding of visual semantics, or does it merely mechanically reduce reliance on textual priors due to the forced redistribution of attention weights? In multimodal large language models (MLLMs), weak grounding is often accompanied by severe hallucination phenomena. The paper lacks counterfactual analysis or visual decoupling experiments, failing to delve into whether this intervention method fundamentally improves the model's ability to truly align visual features.

---

> ### Author Rebuttal · Authors · 2026-03-31
>
> We thank the reviewer for the thorough and constructive feedback. We agree that the paper should distinguish more clearly between what is theoretically established and what is empirically supported, and will revise accordingly.
>
> **W1: Theoretical foundation of the mask prior**
>
> We agree that mask token trajectories are nonlinear and context-dependent. Our claim is much narrower: there exists a dominant, empirically observable low-dimensional bias in the final hidden-state space, and suppressing it improves generation quality. The Linear Representation Hypothesis [1] suggests that even in nonlinear models, meaningful concepts can emerge as approximately linear directions in representation space. Motivated by this, we construct a low-dimensional prior subspace via PCA and suppress aligned components through linear projection. We do not claim that this fully explains entropy collapse — rather, it captures an important, practically useful component. The table below supports the existence of such a dominant linear bias:
>
> | Direction A | Direction B | Cosine Sim |
> |:-----------:|:-----------:|:----------:|
> | $e_{\text{uniform}}$ | $e_{\text{freq}}$ | +0.897 |
> | $e_{\text{uniform}}$ | $e_{\text{mask}}$ | +0.883 |
> | $e_{\text{freq}}$ | $e_{\text{mask}}$ | +0.735 |
> | $e_{\text{random}}$ | $e_{\text{mask}}$ | +0.427 |
>
> The vocabulary mean aligns strongly with mask token hidden states (0.883) while a random direction does not (0.427), justifying linear projection as a targeted suppression mechanism. We will revise the paper to make this scope clearer.
>
> **W2: Potential distortion of 2D visual spatial structure by MRS**
>
> Our claim is not that MRS formally guarantees preservation of 2D visual geometry. MRS rescales RoPE frequency components without changing token indices, patch layout, or sequence order. While prior RoPE rescaling methods suggest frequency adjustment does not inherently destroy positional reasoning, we agree this alone does not establish 2D structural preservation. Our main evidence is empirical:
>
> | Backbone | Variant | RefCOCOg | Ferret | DetailCaps |
> |:--------:|:-------:|:--------:|:------:|:----------:|
> | LLaDA-V | baseline | 64.8 | 60.4 | 59.8 |
> |  | vis only | **65.2** | 62.5 | 59.6 |
> |  | txt only | 65.0 | **63.2** | 60.2 |
> |  | gen only | 64.9 | 61.8 | 60.1 |
> |  | ours (all) | 65.0 | 62.9 | **63.6** |
> | LaViDa | baseline | 36.9 | 25.9 | 8.3 |
> |  | vis only | 43.8 | 35.9 | **56.4** |
> |  | txt only | 43.8 | **36.1** | **56.4** |
> |  | gen only | 43.8 | 35.8 | **56.4** |
> |  | ours (all) | **44.0** | 35.7 | 56.1 |
>
> Visual-only MRS improves grounding on both backbones (LLaDA-V: +0.4/+2.1; LaViDa: +6.9 on RefCOCOg), suggesting MRS does not damage visual spatial structure. We agree 2D-aware RoPE analysis is valuable future work.
>
> **W3: Potential interference with learned diffusion dynamics**
>
> MPS modifies only $k=3$ out of $d=4096$ dimensions (0.073%) — comparable to ITI [3] and SADI [2], which intervene in hidden representations without retraining. We do not claim a formal OOD guarantee. Consistent gains across multiple architectures and tasks argue against broad degradation. We will state this limitation more explicitly.
>
> **W4: Insufficient decoupling of grounding and hallucination**
>
> We agree our experiments do not fully decouple grounding from hallucination causally. Prior work links visual attention to hallucination: OPERA [5] shows hallucinations arise from insufficient visual attention, PAINT [6] shows imbalanced visual attention induces hallucinations, and M3ID [4] shows strengthening visual grounding reduces hallucinations without retraining. MRS is designed in this spirit — Fig. 7(b) confirms MRS increases attention from generation tokens to distant visual tokens.
>
> CHAIR (COCO val, 500 images):
>
> | Method  | CHAIR_s↓ | CHAIR_i↓ |
> |---------|----------|----------|
> | LLaDA-V | 29.4     | 9.5      |
> | + Ours  | **27.0** | **8.3**  |
>
> AMBER-G (1,004 images):
>
> | Method  | CHAIR↓  | Cover↑ | HAL↓     |
> |---------|---------|--------|----------|
> | LLaDA-V | 7.2     | 61.6   | 41.1     |
> | + Ours  | **6.9** | 61.6   | **40.6** |
>
> Hallucination decreases while coverage is maintained, arguing against trivial suppression. We will frame these findings as supportive evidence rather than definitive proof.
>
> **References**
> [1] Park et al., The Linear Representation Hypothesis and the Geometry of Large Language Models, ICML 2024.[2] Wang et al., Semantics-Adaptive Activation Intervention for LLMs via Dynamic Steering Vectors, ICLR 2025. [3] Li et al., Inference-Time Intervention: Eliciting Truthful Answers from a Language Model, NeurIPS 2023.[4] Favero et al., Multi-Modal Hallucination Control by Visual Information Grounding, CVPR 2024.
> [5] Huang et al., OPERA: Alleviating Hallucination in Multi-Modal Large Language Models via Over-Trust Penalty and Retrospection-Allocation, CVPR 2024.
> [6] Arif et al., PAINT: Paying Attention to INformed Tokens to Mitigate Hallucination in Large Vision-Language Model, arXiv 2025.

---

> > ### Author Rebuttal · Reviewer_yd9x · 2026-04-01
> >
> > Thank the authors for the detailed response; I am very satisfied with the reply, it has resolved most of my questions. Therefore, I raised my rating to 5: accept.
> >
> > The authors' perspective on utilizing PCA for exploration is quite interesting. In fact, [a] similarly utilizes PCA to explore how the visual encoder "captures an important, practically useful component" that impacts downstream tasks. This represents a clever approach to controlling and investigating the internal information flow within the model. I hope the authors will consider incorporating this discussion into the final version of the paper to further enrich its contribution to the community.
> >
> > [a] Visual Instance-aware Prompt Tuning. ACM MM 25

---

### Official Review · Reviewer_YPAp · 2026-03-12

**Soundness:** 2
**Presentation:** 3
**Significance:** 3
**Originality:** 3
**Overall Recommendation:** 4
**Confidence:** 3

**Summary:**

This paper investigates two failure modes in diffusion-based vision-language models (VLMs): (1) repetitive generation caused by representation drift of masked tokens toward a shared language prior, and (2) degraded visual grounding due to attention being dominated by nearby mask tokens in the suffix during bidirectional decoding. To address these issues, the authors propose two inference-time interventions. First, Mask Prior Suppression (MPS) performs geometric debiasing on hidden states by suppressing components aligned with a vocabulary-level prior subspace, mitigating repetition caused by mask token collapse. Second, Monotonic RoPE Scaling (MRS) modifies positional encoding frequencies to strengthen long-raStrengthsnge attention, improving interactions between generated tokens and distant visual tokens. The approach is training-free and can be integrated into existing diffusion VLMs, leading to improvements in long-form generation quality and multimodal grounding.

**Compliance With Llm Reviewing Policy:**

Affirmed.

**Final Justification:**

The authors have addressed most of my questions during the rebuttal stage. I find the motivation of the paper reasonable, and the proposed method brings a certain degree of improvement. However, considering that the gains are relatively limited and the applicability is mainly restricted to full-suffix models, I am inclined to maintain a weak accept rating.

**Key Questions For Authors:**

same as Weakness.

**Limitations:**

The proposed solutions are empirical inference-time interventions tailored to the current full-suffix diffusion decoding setup. While the results are promising, it remains unclear whether the same mechanisms apply to more efficient diffusion inference paradigms, such as block-wise or semi-autoregressive diffusion models. Evaluating the methods under these alternative decoding frameworks would help clarify the generality of the proposed insights.

**Strengths And Weaknesses:**

Strengths:
- The paper provides a useful analysis of two practical issues in diffusion-based decoding: representation collapse of masked tokens and positional attention bias toward nearby mask tokens.
- Both proposed methods (MPS and MRS) are lightweight and training-free, making them easy to integrate into existing models.

Weaknesses
- MPS assumes that the collapse direction of masked token representations aligns with the mean of vocabulary embeddings. However, the average of token embeddings does not necessarily correspond to a meaningful semantic prior in discrete language spaces. It is unclear why this specific direction should represent the dominant collapse direction rather than alternatives such as mask embedding propagation. Additional empirical evidence comparing different candidate priors would strengthen the claim.
- The paper attributes degraded grounding to large numbers of suffix mask tokens participating in attention. However, recent diffusion language model work such as DPad (suffix dropout) and D2F (block-wise AR–diffusion decoding) suggests that this issue can also be mitigated by redesigning the decoding process to reduce or eliminate redundant suffix mask tokens. This raises the question of whether the proposed positional scaling addresses a fundamental limitation of diffusion decoding or compensates for a particular implementation choice. It would be useful to compare MRS with simpler strategies such as dropping redundant suffix mask tokens or adopting block-wise decoding structures.

---

> ### Author Rebuttal · Authors · 2026-03-31
>
> We thank the reviewer for the thoughtful and constructive feedback.
>
> **W1: Why is the vocabulary mean a reasonable prior direction?**
>
> Our claim is intentionally empirical and limited in scope: we use the vocabulary mean as a simple, stable proxy for a shared bias direction in the final hidden state space. We do **not** claim that it fully characterizes the nonlinear dynamics of mask-token collapse. Rather, our claim is that it captures an important and empirically observable component of that collapse.
>
> To test this directly, we compared several candidate prior directions in the final hidden state space, including a frequency-weighted lexical prior estimated from COCO train2014 caption frequencies:
>
> | Direction A | Direction B | Cosine Sim |
> |:-----------:|:-----------:|:----------:|
> | $e_{\text{uniform}}$ | $e_{\text{freq}}$ | +0.897 |
> | $e_{\text{uniform}}$ | $e_{\text{mask}}$ | +0.883 |
> | $e_{\text{freq}}$ | $e_{\text{mask}}$ | +0.735 |
> | $e_{\text{random}}$ | $e_{\text{mask}}$ | +0.427 |
>
> Although $e_{\text{uniform}}$ does not explicitly encode corpus frequency, it is highly aligned with $e_{\text{freq}}$ in practice ($0.897$), indicating that both capture a very similar dominant lexical direction in the final hidden state space. Moreover, both $e_{\text{uniform}}$ and $e_{\text{freq}}$ are substantially more aligned with mask-token hidden states than a random direction ($0.883$ and $0.735$ vs. $0.427$), supporting the view that the vocabulary mean is a practical proxy for a shared collapse direction rather than an arbitrary choice.
>
> We also compared the vocabulary mean against top-$k$ token-based priors (Table 3), where the vocabulary mean yields consistently stronger performance. In addition, Fig. 2(b) shows that the propagated mask-token trajectory and the vocabulary-mean trajectory converge to a similar region in the final layer. We therefore use the vocabulary mean not because it is fundamentally different from a propagated mask prior, but because it provides a simple, parameter-free, and stable estimate of a closely related shared bias component in practice.
>
> **W2: Comparison with existing positional and structural baselines**
>
> We agree that structural changes to the decoding process are an important related direction.
>
> We therefore added results with a DPad-style suffix-dropping baseline. This comparison is particularly relevant because DPad is training-free and directly modifies suffix attention during inference, making it straightforward to adapt to existing LDVLM checkpoints:
>
> | Method        | RefCOCOg | Ferret   | DetailCaps |
> |---------------|----------|----------|------------|
> | LLaDA-V       | 64.8     | 60.4     | 59.8       |
> | + DPad        | 64.7     | 60.3     | 60.2       |
> | + Ours        | **65.0** | **62.9** | **63.6**   |
> | + DPad + Ours | **65.0** | 61.7     | 60.5       |
> | LaViDa        | 36.9     | 25.9     | 8.3        |
> | + DPad        | 39.6     | 30.8     | 7.9        |
> | + Ours        | 44.0     | 35.7     | 56.1       |
> | + DPad + Ours | **44.4** | **36.1** | **56.6**   |
>
> Our method consistently outperforms DPad alone across both backbones. The two methods address related but distinct mechanisms: DPad reduces redundant suffix-mask influence at the decoding/input level, whereas MRS strengthens long-range access to informative visual tokens at the representation level. Their combination is largely complementary on LaViDa. On LLaDA-V, DPad's input-level masking may partially remove positional context that MRS relies on, which is why the two methods are not always additive in that setting.
>
> Regarding D2F, we view it as an important related direction, but not as a directly comparable plug in baseline in our setting. D2F changes the decoding paradigm into a block wise autoregressive and diffusion hybrid and is realized through asymmetric distillation from a pretrained dLLM, rather than as a training free inference time modification. Moreover, it is introduced for text dLLMs, so adapting it to existing LDVLM checkpoints would likely require additional multimodal adaptation or training rather than direct insertion into our current models.
>
> **Limitations**
>
> Our current experiments focus on the full suffix diffusion decoding setup used by current LDVLMs. At the same time, we do not view MPS or MRS as inherently tied to that exact implementation. The DPad results already suggest that our methods remain meaningful even when suffix redundancy is partially reduced. More broadly, we believe these interventions are potentially compatible with more efficient decoding paradigms, including semi autoregressive or block wise diffusion variants. However, validating this fully, especially for methods such as D2F that require a different decoding architecture and additional training, would require new multimodal adaptation and is an important direction for future work.

---

> > ### Author Rebuttal · Reviewer_YPAp · 2026-04-01
> >
> > The authors have addressed most of my questions and concerns. However, considering that the improvements over methods such as DPad do not appear to be substantial, and given the known limitations, I am inclined to maintain my current rating.

---

### Official Review · Reviewer_W2x1 · 2026-03-21

**Soundness:** 2
**Presentation:** 3
**Significance:** 3
**Originality:** 2
**Overall Recommendation:** 3
**Confidence:** 4

**Summary:**

This paper identifies two primary failure modes in long-form multimodal generation for Large Diffusion Vision-Language Models (LDVLMs): mask prior drift leading to repetitive token emission and positional attention collapse caused by RoPE’s locality bias.To mitigate these issues, the authors introduce two training-free, test-time techniques: Mask Prior Suppression (MPS) to remove biased components from hidden states and Monotonic RoPE Scaling (MRS) to stabilize long-range visual grounding.Experimental results across various backbones and benchmarks demonstrate that the proposed methods effectively enhance lexical diversity, reduce redundancy, and improve fine-grained visual alignment in long-form captioning tasks.

**Compliance With Llm Reviewing Policy:**

Affirmed.

**Key Questions For Authors:**

1.Can you report results with established captioning metrics (e.g., CIDEr/SPICE/RefCLIPScore/HALO) on larger subsets to corroborate the distinct-n/repetition improvements, especially for DetailCaps and LLaVA-Bench?
2.Does MPS applied at intermediate layers (or per-layer with smaller strength) yield further benefits, or is the final-layer-only intervention strictly better in latency/quality trade-offs?
3.How does MRS interact with segment structure?

**Limitations:**

Yes

**Strengths And Weaknesses:**

Strengths：
1. The paper provides multiple lines of evidence: distinct-n/repetition analyses vs. generation steps, cosine similarity trends of contextualized masks to vocabulary mean, and frequency-wise attention decomposition over relative distances.
2. MPS is a simple, principled linear intervention: constructing a small prior subspace via PCA of layer-wise forward passes of the vocabulary-mean embedding and attenuating aligned components at inference. This is a neat adaptation of debiasing/projection ideas to diffusion text decoding.


Weakness：
1.The prior subspace is constructed from single-token forward passes of the vocabulary mean through all layers, which may not faithfully reflect contextualized dynamics under realistic multimodal inputs and bidirectional attention.
2.Missing comparisons to known RoPE scaling baselines (NTK-aware/YaRN/LongRoPE) adapted to the LDVLM setting.
3. Some evaluations rely on small sampled subsets.
4.There are minor equation and notation inconsistencies.
5.It is unclear whether MRS applies uniformly to all token segments and whether segment-specific scaling was considered.

---

> ### Author Rebuttal · Authors · 2026-03-31
>
> We thank the reviewer for the detailed and
> constructive feedback. We have added
> representative baseline comparisons and
> additional ablations, and will correct the
> notation inconsistencies in the final version.
>
> **W1: Context-free prior estimation**
>
> We agree that a context-free prior is only
> an approximation to the full contextualized
> dynamics. Our claim is limited: a dominant
> low-dimensional bias component is
> consistently present in the final hidden
> state space, and suppressing it improves
> generation quality. To verify this, we
> measured the cosine similarity between the
> context-free prior direction and
> contextualized mask-token hidden states:
>
> | T          | 32    | 16    | 8     | 4     |
> |------------|-------|-------|-------|-------|
> | Cosine sim | 0.255 | 0.256 | 0.268 | 0.283 |
>
> The alignment remains stable throughout
> decoding, while random directions are
> substantially less aligned. We will clarify
> this empirical scope in the revision.
>
> **W2: Comparison with representative RoPE
> scaling baselines**
>
> | Method  | RefCOCOg | Ferret   | DetailCaps |
> |---------|----------|----------|------------|
> | LLaDA-V | 64.8     | 60.4     | 59.8       |
> | + NTK   | 61.9     | 59.6     | 60.9       |
> | + YaRN  | 59.3     | 56.0     | 59.1       |
> | + Ours  | **65.0** | **62.9** | **63.6**   |
> | LaViDa  | 36.9     | 25.9     | 8.3        |
> | + NTK   | 39.6     | 30.9     | 7.9        |
> | + YaRN  | 39.6     | 30.7     | 7.9        |
> | + Ours  | **44.0** | **35.7** | **56.1**   |
>
> These baselines were designed for
> long-context extrapolation, whereas the
> bottleneck in our setting is RoPE's
> locality bias under full-suffix iterative
> unmasking. Simply reducing positional
> frequencies does not address over-attention
> to nearby but semantically weak mask
> tokens. In contrast, MRS monotonically
> amplifies low-frequency components,
> improving long-range visual grounding
> while preserving token ordering.
>
> **W3 & Q1: Additional reference-based metrics**
>
> We report reference-based metrics on
> RefCOCOg (7,573 images), which provides
> human-annotated ground-truth references and
> is therefore well suited for standard
> captioning metrics.
>
> | Method  | CIDEr    | BLEU-4  | METEOR   | RefCLIPScore |
> |---------|----------|---------|----------|--------------|
> | LLaDA-V | 64.8     | 8.2     | 14.6     | 72.8         |
> | + Ours  | **65.0** | **8.4** | 14.6     | 72.8         |
> | LaViDa  | 36.9     | 1.5     | 8.4      | 68.1         |
> | + Ours  | **44.0** | **4.9** | **11.3** | **69.4**     |
>
> For DetailCaps and LLaVA-Bench, standard
> reference-based metrics are less directly
> applicable: DetailCaps does not provide
> canonical reference captions, and
> LLaVA-Bench uses GPT-4 judge scores as its
> evaluation protocol. We therefore report
> task-level scores as the primary evidence
> for these settings.
>
> **W4: Notation inconsistencies**
>
> We will unify the notation across the
> equations, algorithm description, and main
> text, including the prior subspace,
> projection operator, and token groups.
>
> **Q2: MPS at intermediate layers vs.
> final-layer only**
>
> | Method             | prior | RefCOCOg | LLaVA-B  | DetailCaps |
> |--------------------|-------|----------|----------|------------|
> | LLaDA-V            | —     | 64.8     | 61.3     | 59.8       |
> | + all layers (×32) | 0.003 | 61.3     | 63.5     | 60.2       |
> | + last 8 layers    | 0.013 | **65.1** | 61.1     | 59.8       |
> | + last 3 layers    | 0.033 | 64.8     | 61.1     | 60.3       |
> | + ours (last 1)    | 0.1   | 65.0     | **64.1** | **63.6**   |
> | LaViDa             | —     | 36.9     | 39.5     | 8.3        |
> | + all layers (×32) | 0.009 | 40.0     | 37.4     | 6.7        |
> | + last 8 layers    | 0.038 | 39.6     | 36.7     | 8.1        |
> | + last 3 layers    | 0.1   | 39.7     | 35.8     | 8.1        |
> | + ours (last 1)    | 0.3   | **44.0** | **46.5** | **56.1**   |
>
> Final-layer-only provides the strongest
> overall trade-off and is consistently best
> on long-form settings. Applying suppression
> too early can interfere with intermediate
> representations that still encode useful
> context.
>
> **W5 & Q3: Segment-specific application
> of MRS**
>
> Please also refer to our response to
> Reviewer yd9x (W2), which provides the full
> segment-wise ablation table, including
> LaViDa results. In summary, no single
> segment consistently dominates, and the
> uniform all-token setting provides the
> strongest or near-strongest overall
> performance without task-specific tuning.

---

> > ### Author Rebuttal · Reviewer_W2x1 · 2026-04-01
> >
> > In response to the author's rebuttal, I stand by my original score.

---

### Decision · Program_Chairs · 2026-04-30

**Decision:**

Accept (regular)

**Comment:**

Given the paper’s empirical contributions and practical value, and the fact that the authors effectively addressed concerns from two reviewers in their rebuttal, this method offers actionable insights for long-text multimodal generation. It is likely to be well-received by the community. However, due to limited theoretical grounding and generalizability, my recommendation is 'Weak Accept'.